# Plasmacytoid dendritic cells control dengue and Chikungunya virus infections via IRF7-regulated interferon responses

Brian Webster[1][†], Scott W Werneke[2,3][†], Biljana Zafirova[2], Sébastien This[1], Séverin Coléon[1], Elodie Décembre[1], Helena Paidassi[1], Isabelle Bouvier[2], Pierre-Emmanuel Joubert[2], Darragh Duffy[2], Thierry Walzer[1], Matthew L Albert[2,3]*, Marlène Dreux[1]*

[1]CIRI, Inserm, U1111, Université Claude Bernard Lyon 1, CNRS, UMR5308, École Normale Supérieure de Lyon, Univ Lyon, Lyon, France; [2]Immunobiology of Dendritic Cells, Institut Pasteur, Paris, France; [3]Cancer Immunology Department, Genentech, San Francisco, United States

*For correspondence:
albert.matthew@gene.com (MLA);
marlene.dreux@ens-lyon.fr (MD)

[†]These authors contributed equally to this work

Competing interests: The authors declare that no competing interests exist.

**Abstract** Type I interferon (IFN-I) responses are critical for the control of RNA virus infections, however, many viruses, including Dengue (DENV) and Chikungunya (CHIKV) virus, do not directly activate plasmacytoid dendritic cells (pDCs), robust IFN-I producing cells. Herein, we demonstrated that DENV and CHIKV infected cells are sensed by pDCs, indirectly, resulting in selective IRF7 activation and IFN-I production, in the absence of other inflammatory cytokine responses. To elucidate pDC immunomodulatory functions, we developed a mouse model in which IRF7 signaling is restricted to pDC. Despite undetectable levels of IFN-I protein, pDC-restricted IRF7 signaling controlled both viruses and was sufficient to protect mice from lethal CHIKV infection. Early pDC IRF7-signaling resulted in amplification of downstream antiviral responses, including an accelerated natural killer (NK) cell-mediated type II IFN response. These studies revealed the dominant, yet indirect role of pDC IRF7-signaling in directing both type I and II IFN responses during arbovirus infections.
DOI: https://doi.org/10.7554/eLife.34273.001

## Introduction

Upon sensing invading viruses, host cells produce type I interferons (IFNs), leading to the expression of an array of IFN-stimulated genes (ISGs). This first-line response suppresses viral spread by generating an antiviral state within host cells, and supports the initiation of adaptive immunity (*Hoffmann et al., 2015*). Viral sensing may involve non-hematopoietic or hematopoietic cells that are targets of infection. Specific pathogen-associated motifs, such as viral nucleic acids, are recognized by pattern recognition receptors, which can be cytoplasmic (e.g. retinoic inducible gene-I (RIG-I)-like receptors, and NOD-like Receptors) or endosomal (e.g. Toll-like receptors; TLRs) (*Jensen and Thomsen, 2012*). Due to the potency of these innate responses, most viruses have evolved mechanisms to subvert or evade pathogen-sensing pathways (*GarciaGarcía-Sastre, 2017*).

We and others have recently highlighted the existence of alternative or indirect pathogen-sensing mechanisms that circumvents cell-intrinsic viral evasion mechanisms (*Webster et al., 2016*). Such alternative pathways involve the sensing of infected target cells by plasmacytoid dendritic cells (pDCs), a DC subtype specialized in the production of robust level of type I IFNs (referred herein to as IFN-I) (*Swiecki and Colonna, 2015*). Notably, this was illustrated in the context of dengue virus (DENV), a positive-sense single-stranded RNA virus, which represents global health concerns (*Bhatt et al., 2013*). Recent in vitro work highlighted a newly defined mode of pDC activation, which

**eLife digest** Viruses, like the ones responsible for the tropical diseases dengue and chikungunya, are parasites of living cells. As they cannot multiply on their own, these microbes need to infect a host cell and hijack its machinery to make more of themselves.

When a cell is invaded, it can sense the viral particles, and defend itself by releasing antiviral molecules. Some of these molecules, such as interferons, also help recruit immune cells that can fight the germs. However, viruses often evolve mechanisms to escape being detected by the cell they occupy.

Plasmacytoid dendritic cells are a rare group of immune cells, and they are able to detect when another cell is infected by the dengue virus. When they are in close physical contact with an invaded cell, these sentinels can recognize immature viral particles and release large amounts of antiviral molecules. However, it is unclear how important plasmacytoid dendritic cells are in clearing a viral infection.

Here, Webster, Werneke et al. confirmed that plasmacytoid dendritic cells were able to sense cells infected by dengue, but also by chikungunya. When this happened, the dendritic cells primarily produced interferon, rather than other defense molecules.

In addition, mice were genetically engineered so that the production of interferon was restricted to the plasmacytoid dendritic cells. When infected with dengue or chikungunya, the modified rodents resisted the diseases. These results show that, even though they are only a small percentage of all immune cells, plasmacytoid dendritic cells have an outsize role as first responders and as coordinators of the immune response.

Finally, Webster, Werneke et al. showed that when low doses of interferon are added, , the plasmacytoid dendritic cells respond more quickly to cells infected by dengue. Together these findings could potentially be leveraged to create new treatments to fight dengue. These would be of particular interest because interferons are not as damaging to the body compared to other types of defense molecules. The issue is timely since climate change is allowing the mosquitos that transmit dengue and chikungunya to live in new places, exposing more people to these serious infections.

DOI: https://doi.org/10.7554/eLife.34273.002

is mediated primarily by non-infectious immature DENV particles and requires physical cell-cell contact with DENV-infected cells (*Décembre et al., 2014*). Importantly, such cell-cell contact-dependent activation of pDCs has also been reported for several other genetically distant viruses (*Webster et al., 2016*). Consistently, DENV, like some other viruses, does not to directly infect pDCs (*Décembre et al., 2014*; *Webster et al., 2016*). We thus aimed to test how such indirect cell-cell sensing of viral pathogens by pDCs, in the absence of pDC infection, engages host mechanisms for in vivo viral clearance.

pDCs function as sentinels of viral infection, predominantly via recognition of single-stranded RNA and unmethylated CpG containing DNA by endosomal TLR7 and TLR9, respectively. Activation of TLR7 or TLR9 results in copious secretion of IFN-I as well as other proinflammatory cytokines (notably TNFα), driven by the transcription factor(s) interferon regulatory factor (IRF)−7 and nuclear factor-kappa B (NF-κB), respectively (*Swiecki and Colonna, 2015*). Interestingly, despite the inherent ability of pDCs to produce both IFN-I and NF-κB-induced cytokines, previous studies have suggested a 'bifurcated' pattern of TLR7/9 signaling, where IFN-I production may occur in the absence of NF-κB activation, and vice versa (*Swiecki and Colonna, 2015*). The production of IFN-I or proinflammatory cytokines may in part be dependent on the sub-cellular compartment in which these TLRs encounter activating signal, as CpG-A accumulates in early endosomes to induce IFN-I whereas CpG-B aggregates in endolysosomes to activate NF-κB (*Swiecki and Colonna, 2015*). In the context of viral infection, pDCs exposed to cell-associated Hepatitis C virus produced IFN-I but not NF-κB-dependent cytokines (*Dental et al., 2012*). Despite these observations, the consequences of pDC IFN-I production in the absence of NF-κB induction remain unclear, and have not yet been shown to be sufficient for in vivo viral control.

Herein, we show that indirect activation of pDCs by contact with DENV or CHIKV infected cells results in an IRF7-induced IFN response, in the absence of NF-κB inflammatory responses. We developed a novel mouse model where *Irf7* expression is pDC-restricted, that is, *Irf3*$^{-/-}$;*Irf7*$^{-/-}$ double knockout mice, with *Irf7* expression driven under the pDC-specific promoter *Sialic acid binding Ig-like lectin H* (*Siglech*) (*Blasius et al., 2006*; *Takagi et al., 2011*; *Zhang et al., 2006*). We demonstrated that pDC-restricted IRF7-induced signaling is sufficient to achieve in vivo control of DENV and CHIKV infections. We further elucidated that the early pDC IRF7-mediated response accelerates type II IFN (IFNγ) responses via natural killer (NK) cell activation, thus positioning pDCs as a cell type that regulates the interplay between type I and type II IFN responses in the control of these viral infections, an immune axis that is independent of other sources of type I IFN or pDC-derived NF-κB-induced cytokines.

## Results

### Cell-cell sensing of DENV-infected cells does not induce inflammatory NF-κB responses by pDCs

We previously reported that ex vivo activation of an antiviral response by pDCs requires physical contact with DENV-infected cells (*Décembre et al., 2014*). Prior studies suggested that pDC TLR7/9 signaling differentially induced IRF7 and/or NF-κB, depending on the ligand (*Swiecki and Colonna, 2015*). Thus, we determined relative pDC secretion of representative NF-κB-induced (TNFα) and IRF7-induced (IFNα) cytokines in response to DENV. The sensing of DENV-infected cells by human pDCs failed to induce TNFα secretion (*Figure 1A*). This was in contrast to the robust TNFα production triggered by direct TLR7 stimuli, including the synthetic ligand imiquimod (IMQ) and influenza A [Flu], which directly activates pDCs as cell-free virus as previously reported (*Décembre et al., 2014*) (*Figure 1A*).

pDC activation by DENV, the result of cell-cell transfer of TLR7 ligands, may occur more slowly than direct activation by cell-free stimuli. To assess if differential kinetics of pDC activation underlie the difference in TNFα production, we evaluated pre-treatment of human pDCs with low-dose IFNβ, prior to coculture with infected cells. Such pre-treatment regimens have been previously shown to accelerate both IFNα and TNFα production in response to influenza virus (*Phipps-Yonas et al., 2008*). While IFNβ pretreatment accelerated the kinetics of pDC-derived IFNα (all stimuli) and TNFα production (IMQ/Flu), we observed minimal TNFα production in response to DENV-infected cells (*Figure 1B*). These observations imply that the defect in DENV-mediated TNFα induction by pDCs might be due to the indirect mechanism of stimulation. Furthermore, we showed that pDC TNFα levels in response to IMQ were not diminished in presence of DENV-infected cells (*Figure 1—figure supplement 1A*). Together, these results indicated that DENV is neither promoting nor preventing the activation of NF-κB-induced TNFα production by pDCs.

We extended these findings to mouse pDC activation, utilizing cells derived from either bone marrow (BM) or isolated from spleen. We also assessed a broader range of NF-κB-dependent genes. DENV-infected cells induced higher levels of IFNα and interferon-stimulated genes (ISGs) as compared to synthetic TLR7/8 agonists (R848 and IMQ) (*Figure 1C*, *Figure 1—figure supplement 1C–D*), results that were consistent with human pDC findings (*Figure 1A–B*). Conversely, the TLR7/8 agonist R848 induced a strong NF-κB-dependent gene signature in mouse pDCs, whereas limited induction was observed when using DENV-infected cells (*Figure 1D* and *Figure 1—figure supplement 1C*). Together, these results demonstrated that pDC-mediated sensing of DENV-infected cells primarily leads to IFN-I production and a strong ISG response in the absence of NF-κB-mediated inflammatory responses.

### An in vivo model for pDC-restricted IFN-I production

To address how this IRF7- and IFN-I-restricted response to DENV may position pDCs as critical in the control of viral infections, we generated a unique mouse model where IRF7 signaling is restricted to pDCs. We established a knock-in strain where *Irf7* expression is driven by the pDC-specific *Siglech* promoter (thus called *Siglech*$^{Irf7/+}$) (*Figure 2A*). We next backcrossed these mice onto *Irf3*$^{-/-}$;*Irf7*$^{-/-}$ double knockout mice to generate hemizygous *Irf7*-expressing animals (*Siglech*$^{Irf7/+}$;*Irf3*$^{-/-}$;*Irf7*$^{-/-}$ referred to as 'pDC:Irf7$^+$' mice). Use of hemizygous mice preserved one copy of the *Siglech* gene

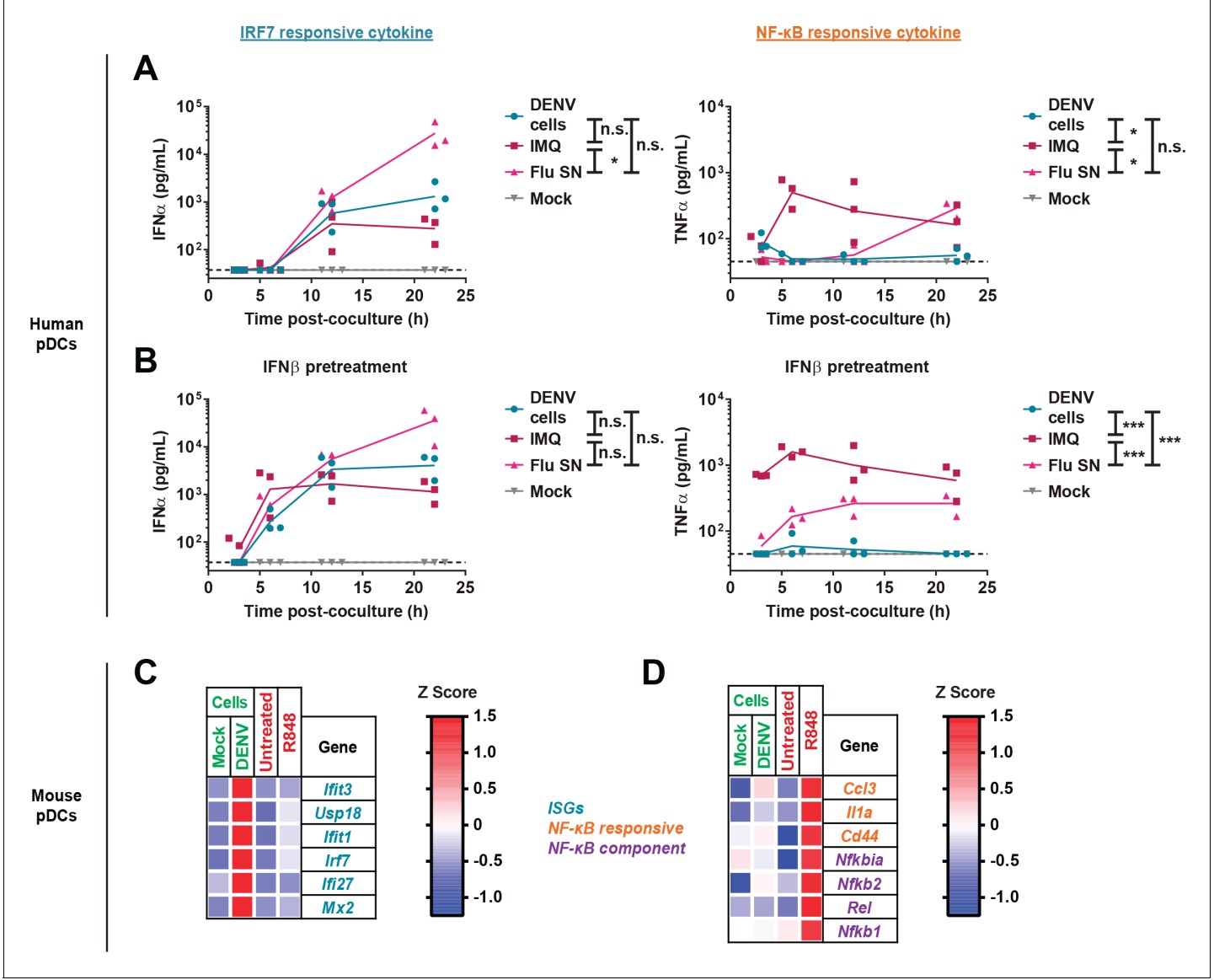

**Figure 1.** Lack of NF-κB response by pDCs in ex vivo coculture with DENV infected cells. (A) Quantification of IFNα and TNFα in supernatants of human pDCs cocultured with DENV-infected (DENV cells) or uninfected BHK-21 cells (Mock), in presence of synthetic TLR7 agonist: Imiquimod; IMQ; 1 µg/mL or influenza infectious supernatant; Flu; $3 \times 10^3$ FFU. (B) pDCs were pretreated for 3 hr with low-dose IFNβ (50 U/mL) and then cocultured or treated as in (A). Median, $n = 3$ independent experiments. (C–D) qRT-PCR analysis of gene expression (using mouse-specific primers) by WT murine splenic pDCs cocultured for 22 hr with uninfected- (mock cells) or DENV-infected Huh7.5.1 cells (DENV cells) or treated with TLR7/8 agonist (R848). Expression levels are normalized to housekeeping genes (β-actin) and shown as Z scores (red-blue gradient from maximum to minimum expression).
DOI: https://doi.org/10.7554/eLife.34273.003

The following figure supplement is available for figure 1:

**Figure supplement 1.** Lack of NF-κB responses to DENV in murine bone marrow and splenic pDCs, and retention of pDC ISG responses in the absence of *Irf3/Irf7*.
DOI: https://doi.org/10.7554/eLife.34273.004

(*Figure 1—figure supplement 1B*). Irf3/7 double knockout mice (referred to as Irf3/7 DKO mice), deficient in IFN-I production (*Rudd et al., 2012*; *Schilte et al., 2012*) were used as comparator negative controls in all experiments.

To validate the *Irf7* knock-in in pDC:Irf7[+] mice, we analyzed IRF7 protein levels in DC subsets. pDCs were the only cell type to retain significant levels of IRF7 protein expression, seen in both pDC:Irf7[+] and WT mice, but not in Irf3/7 DKO mice (*Figure 2B*). To functionally validate the pDC:

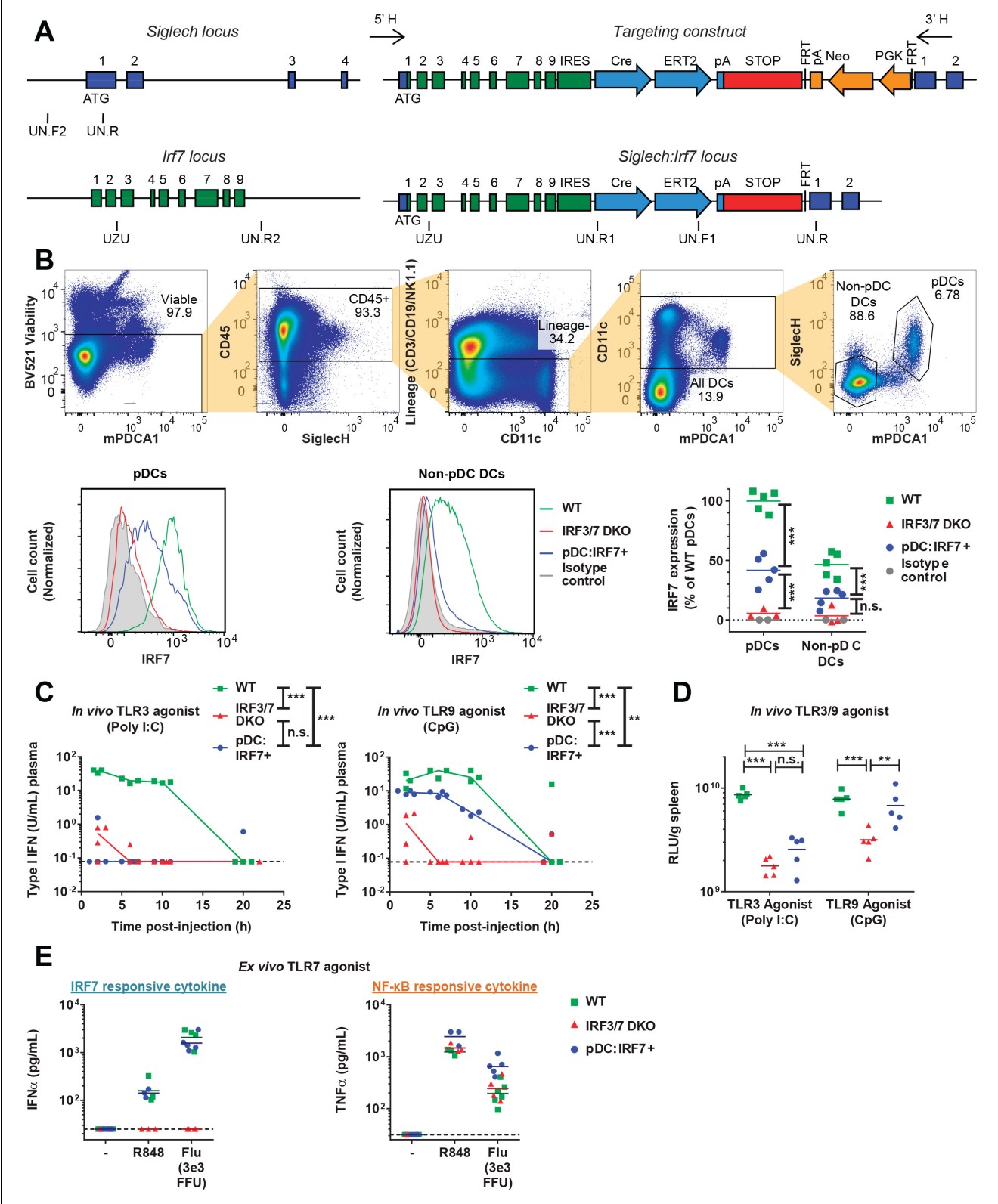

**Figure 2.** Functional validation of the pDC:Irf7+ mouse model. (**A**) Targeting construct for the knock-in of *irf7* under the control of the *siglech* promoter. (**B**) Expression levels of IRF7 analyzed by FACS in pDCs and non-pDC DCs isolated from spleens of uninfected WT, Irf3/7 DKO and pDC:Irf7+ mice. Gating strategy for DCs and pDCs from splenocyte populations (upper panels), IRF7 expression (lower panels); 3–5 mice per condition. (**C–D**) Quantification of IFN activity by bioassay in plasma (**C**) and spleen homogenates (**D**) at various time-points post-injection of mice with agonists of TLR3

*Figure 2 continued on next page*

*Figure 2 continued*

and TLR9, polyinosinic:polycytidylic acid (poly I:C) and CpG-type A oligodeoxynucleotides (CpG), respectively; median, $n$ = 3–5 mice per condition. (E) Quantification of IFNα and TNFα in ex vivo cultures of pDCs (mPDCA1$^+$ cells) isolated from BM of WT, Irf3/7 DKO and pDC:Irf7$^+$ mice and treated for 22 hr with TLR7/8 agonist (R848) or influenza virus infectious supernatant (Flu); median, $n$ = 3–5 mice per condition.
DOI: https://doi.org/10.7554/eLife.34273.005

Irf7$^+$ mice, we assessed IFN-I activity induced upon in vivo treatment with agonists of TLR9 and TLR3, which are expressed or not by pDCs, respectively (*Swiecki and Colonna, 2015*). As expected, we observed IFN-I activity in plasma/spleen of WT mice stimulated by either agonist, whereas little-to-no IFN-I activity was detected in Irf3/7 DKO mice (*Figure 2C–D*). Consistent with the TLR expression patterns in pDCs (*Swiecki and Colonna, 2015*), pDC:Irf7$^+$ mice produced high levels of IFN-I in response to TLR9, but not TLR3 agonists.

Using this model system, we assessed how pDC IRF7-signaling mediates antiviral responses to DENV. First, we purified pDCs from WT, Irf3/7 DKO and pDC:Irf7$^+$ mice, and treated them with TLR7 agonists (R848/IMQ/cell-free Flu) or DENV-infected cells. pDC:Irf7$^+$ and WT pDCs produced similar amounts of IFNα (*Figures 2E* and *3A*), confirming the functionality of IRF7 signaling in pDC:Irf7$^+$ mice. We also tested NF-κB-signaling in pDCs from Irf3/7 DKO and pDC:Irf7$^+$ mice induced by the same TLR7 agonists. Confirming independent activation of NF-κB, we observed TNFα secretion levels in both strains to be comparable to WT mice (*Figures 2F* and *3B*). Of note, ISGs previously defined as IRF5-dependent (e.g. *Mx1, Ifit1*) (*Lazear et al., 2013*) were still upregulated in Irf3/7 DKO pDCs when cocultured with DENV-infected cells or stimulated by other TLR7 agonists (*Figure 1—figure supplement 1D*).

## pDC-IRF7-induced potent downstream ISG responses in absence of detectable IFN-I

To determine whether IFN-I response to viral stimuli was restored in pDC:Irf7$^+$ mice, animals were infected with DENV systemically (intravenously, i.v.), and IFNα/β expression was assessed. High levels of IFNα were detected in both the spleen and plasma of infected WT mice (*Figure 3C–D*), but not Irf3/7 DKO mice, in agreement with previous results (*Chen et al., 2013*). IFNβ levels displayed the same pattern of expression (data not shown). pDC:Irf7$^+$ mice had undetectable IFNα/β levels in both the plasma and spleen at all analyzed times post-infection (p.i.) (*Figure 3C–D*).

We considered the possibility that the pDC response to DENV may occur in a localized manner, with insufficient type I IFN in plasma or spleen to allow its detection. We thus assessed downstream responses of IFNα/β receptor (IFNAR) signaling via qRT-PCR analysis of ISGs in different tissues of DENV-infected mice. The ISGs (*Ifit1, Usp18,* and *Ifit3*) were selected for their robust expression following by IFN-I stimulation, as compared to type II IFN (*Liu et al., 2012*). Importantly, despite undetectable IFNα/β in DENV-infected pDC:Irf7$^+$ mice, an early induction (i.e. 18 h p.i.) of *ISG* RNAs was observed in all tissues analyzed. By contrast, this early induction was not observed in Irf3/7 DKO mice (*Figure 3E*). Of note, basal *ISG* expression levels were lower in Irf3/7 DKO and pDC:Irf7$^+$ mice compared to WT, likely reflecting diminished basal IFNAR signaling in the absence of *Irf3* and *Irf7* (*Gough et al., 2012*). These results indicate that restricted expression of *Irf7* in pDCs is sufficient to generate a systemic ISG response.

## In vivo activation of pDC IRF7 antiviral signaling is sufficient to control DENV infection

The pDC:Irf7$^+$ model provided a means to distinguish, in vivo, the role of pDCs in regulating antiviral IFN-I-mediated versus NF-κB-induced response(s) to DENV infection. We therefore measured IRF/STAT-dependent *ISG* expression in multiple organs, and a larger panel of ISGs and NF-κB-dependent transcripts in the liver of DENV-infected animals. DENV infection induced *ISGs* and NF-κB-dependent transcripts in WT, but not Irf3/7 DKO mice (*Figure 3E–G*; expressed as magnitude-independent Z score and *Figure 3—figure supplement 1*; expressed as magnitude-dependent fold-change), confirming previous findings (*Chen et al., 2013*). Strikingly, despite a strong upregulation of ISGs in DENV-infected pDC:Irf7$^+$ mice, we observed minimal induction of NF-κB-dependent inflammation-related transcripts (*Figure 3F–G* and *Figure 3—figure supplement 1*). Based on these

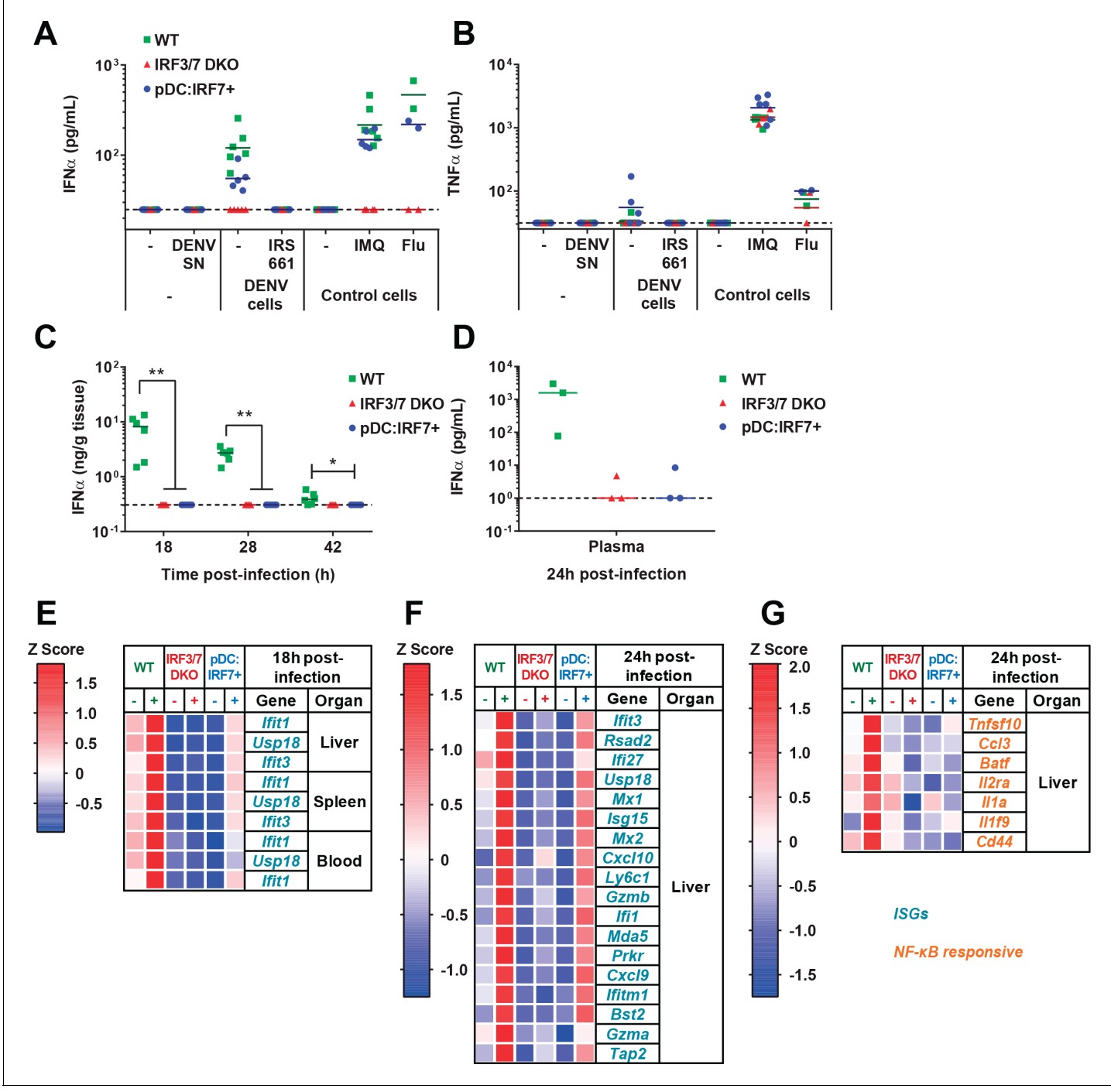

**Figure 3.** pDCs respond in vivo to DENV in a TLR7/IRF7 but not an NF-κB-dependent manner. (**A–B**) Quantification of IFNα (**A**) and TNFα (**B**) by ELISA in ex vivo culture supernatants of pDCs isolated from BM of WT, Irf3/7 DKO and pDC:Irf7⁺ mice and treated for 22 hr with DENV infectious supernatant (DENV SN), TLR7 agonist (IMQ and flu, 3 × 10² ffu), or cocultured with DENV-infected Huh7.5.1 cells ± TLR7 inhibitor IRS661; median, n = 3–5 independent experiments. (**C–G**) Intravenous (i.v.) DENV infection followed by the analysis of IFNα and gene expression in organs collected at the indicated time points p.i. (**C–D**) Quantification of IFNα in spleen homogenates and plasma by ELISA; median; each data point corresponds to an individual mouse: n = 5–6 and n = 3 mice per condition, for spleen and plasma samples, respectively. IFNα was undetectable in uninfected control mice. (**E–G**) qRT-PCR analysis of gene expression in the indicated tissues at 18 hr (**E**) and 24 hr (**F–G**) p.i., and normalized to housekeeping panel (*hprt1*, *β-actin*, *18S*). Expression levels shown as Z scores, n = 2 (**E**) or 3 (**F–G**) infected and n = 1 (**E**) or 2 (**F–G**) uninfected mice per genotype.

DOI: https://doi.org/10.7554/eLife.34273.006

The following figure supplement is available for figure 3:

*Figure 3 continued*

**Figure supplement 1.** Strong early ISG but not NF-κB response to DENV occurs in organs of pDC:Irf7[+]mice.

DOI: https://doi.org/10.7554/eLife.34273.007

in vivo and ex vivo observations, we concluded that the principal signaling pathway downstream of DENV-activated pDCs is IRF7-dependent induction of the IFN-I pathway, which could be wholly segregated from the activation of NF-κB-induced cytokines.

We next sought to determine whether pDC IRF7-induced responses are sufficient to protect mice from DENV. Mice were infected i.v. by DENV and viral RNA and titers were assessed in the spleen, the primary site of replication in infected mice. Viral titer and RNA levels were lower in WT mice than either Irf3/7 DKO or pDC:Irf7[+] mice (*Figure 4* and *Figure 4—figure supplement 1A*), likely due to the higher IFN-I response. Notably, at 42–72 hr post-infection, pDC:Irf7[+] mice displayed significantly reduced levels of DENV titer and RNA in the spleen and plasma compared to Irf3/7 DKO mice (*Figure 4* and *Figure 4—figure supplement 1A*). The higher DENV levels in pDC:Irf7[+] mice as compared to Irf3/7 DKO mice at early time points (i.e. 28 h p.i.) may reflect an enhanced recruitment of DENV-susceptible cells in the spleen of pDC:Irf7[+] mice due to the localized pDC-induced IFN-I response (*Schmid and Harris, 2014*; *Seo et al., 2011*). Our results indicate that IRF7 expression in pDCs is sufficient to control DENV through the local production of IFN-I.

### Type I and type II IFN-induced ISG responses in pDC:Irf7[+] mice

To examine the downstream effector mechanisms initiated by pDC-derived IFN-I, we considered the relative contribution of lymphocyte-derived type II IFN (IFN-II) to viral clearance. Notably, an IFN-II response has been observed even in the absence of *Irf3* and *Irf7* expression in response to DENV infections, and was demonstrated to eventually control DENV infection (*Chen et al., 2013*). To assess IFN-II responses, we measured *Ifng* and *Gbp2* expression, the latter being known to specifically define an IFN-II response (*Hall et al., 2012*). Interestingly, *Ifng* and *Gbp2* up-regulation

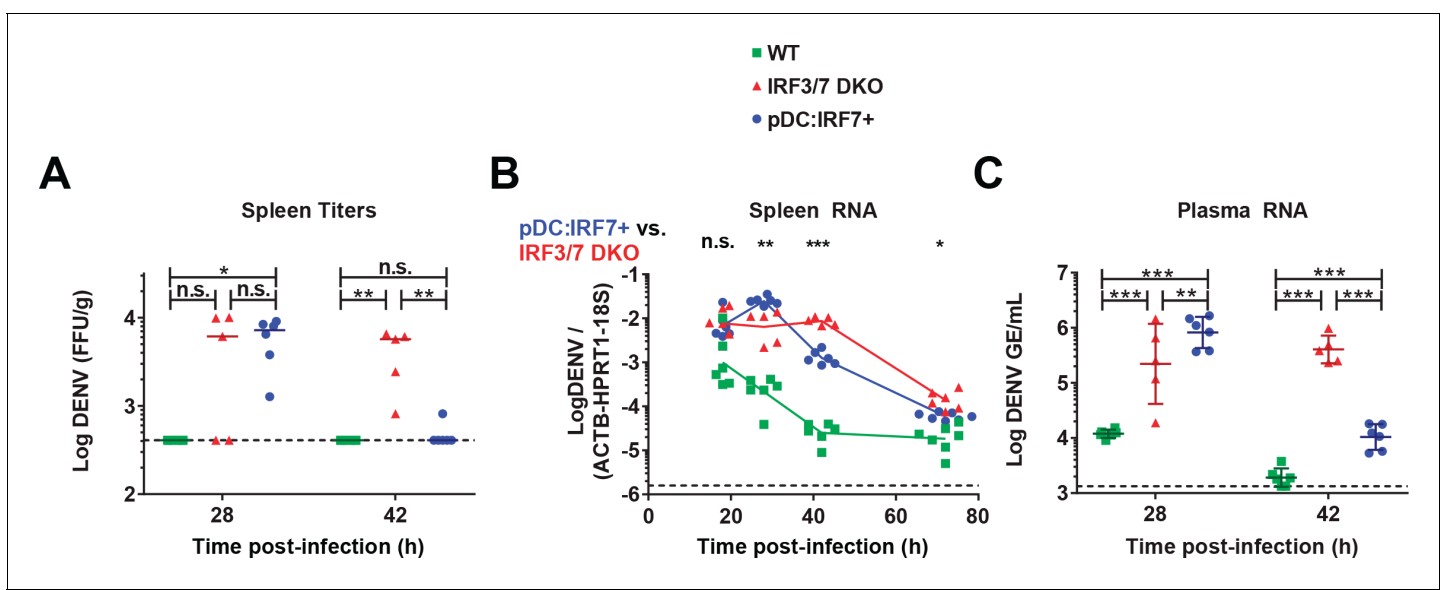

**Figure 4.** pDC:Irf7[+]mice control DENV viremia. DENV infection (i.v.) of mice with the indicated genotypes followed by the analysis of DENV titers/RNA in spleens (A–B) and plasma (C) collected at the indicated time points p.i. DENV RNA levels are normalized to a panel of housekeeping genes (*hprt1*, *β-actin*, and *18S* rRNA) and to exogenous RNA (*xef1A*) for spleen and plasma samples, respectively. Results are expressed as normalized Log₁₀ DENV genome equivalents (GE) or Log₁₀ foci forming unit (ffu)/g tissue; RNA: mean ±SD, titers: median, *n* = 5–7 mice (spleen), *n* = 5–6 mice (plasma).

DOI: https://doi.org/10.7554/eLife.34273.008

The following figure supplement is available for figure 4:

**Figure supplement 1.** Regulation of splenic pDC markers and DENV levels is concomitant with IFN responses.

DOI: https://doi.org/10.7554/eLife.34273.009

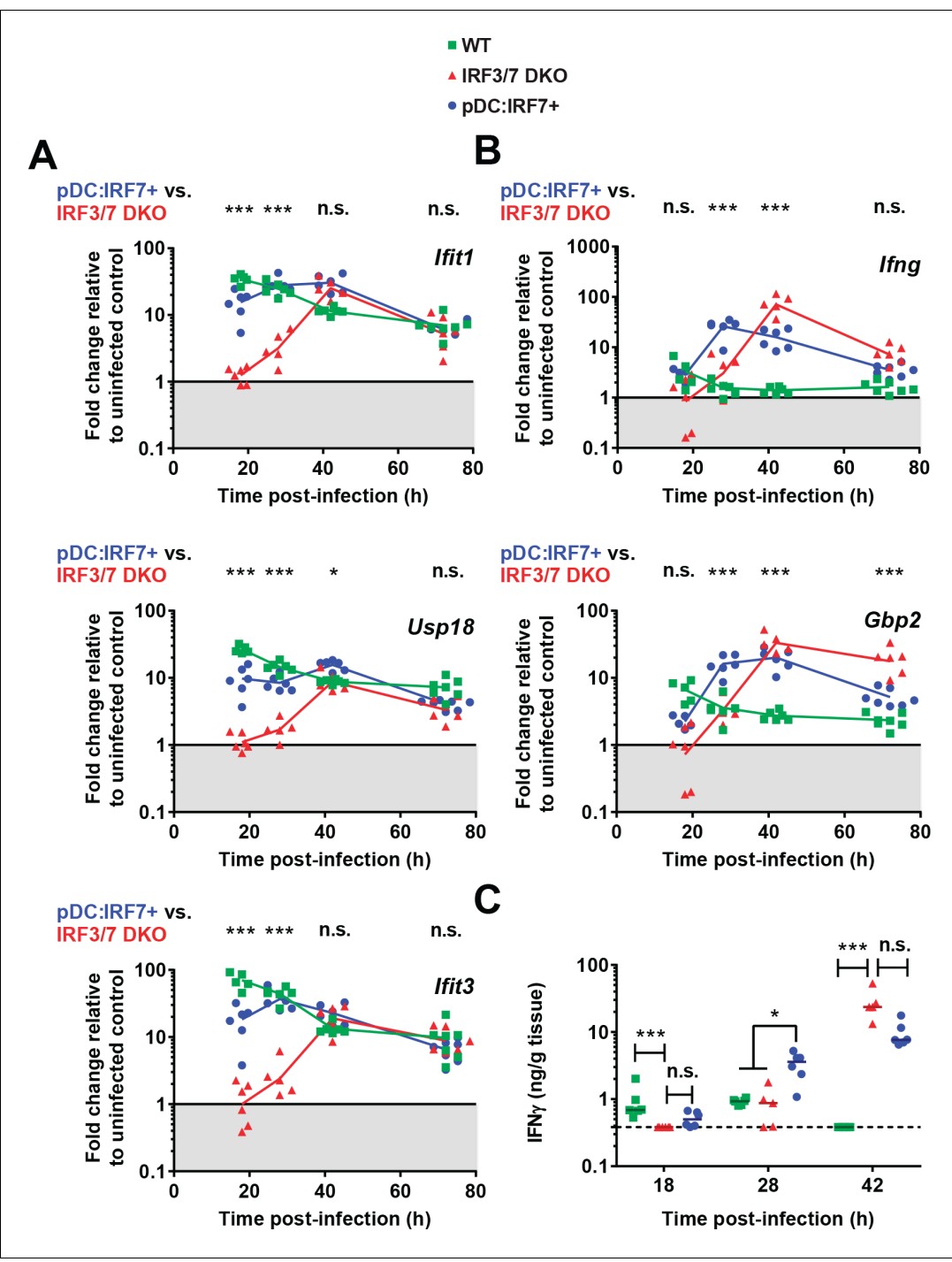

**Figure 5.** pDC:Irf7+mice display elevated IFN-I and accelerated IFN-II responses compared to Irf3/7 DKO mice during DENV infection. Mice were infected as in *Figure 4*. (A–B) Type I IFN-induced ISGs (A) and ISG specifically induced by type II IFN signaling (B). RNA levels were assessed in the spleen and normalized to housekeeping panel (*hprt1*, *β-actin*, *18S*). Transcript levels expressed as the fold change relative to uninfected mice for each genotype; geometric mean, *n* = 5–7 mice per condition, each data point corresponds to an individual mouse. (C) Quantification of IFNγ in spleen homogenates by ELISA; median, *n* = 5–6 mice per condition. IFNγ was undetectable in uninfected control mice.

DOI: https://doi.org/10.7554/eLife.34273.010

The following figure supplement is available for figure 5:

**Figure supplement 1.** Protection in pDC:Irf7+mice involves IFNAR1.

*Figure 5 continued on next page*

*Figure 5 continued*

DOI: https://doi.org/10.7554/eLife.34273.011

(*Figure 5B–C*) temporally correlated with the induction of IFN-I-induced ISGs in DENV-infected Irf3/7 DKO mice (*Figure 5A* and *Figure 4—figure supplement 1B–E*). Of note, these ISGs including *Ifit1/Usp18/Ifit3* respond more highly to IFN-I, but will still be induced to some extent by type II IFN [(*Chen et al., 2013*; *Hall et al., 2012*; *Liu et al., 2012*) and data not shown]. Therefore, in agreement with these prior studies, IFN-II signaling likely mediates the late induction of a larger set of ISGs observed in Irf3$^{-/-}$/7$^{-/-}$ mice.

The kinetic profile of the type II IFN response was markedly different in pDC:Irf7 +mice as compared to Irf3/7 DKO mice, with *Ifng* and *Gbp2* expression detected as early as 18 h p.i and peaking at 28 h p.i. (*Figure 5B–C*). This led to the compelling hypothesis that early pDC IRF7-mediated signaling accelerates the IFN-II response, which is known to be part of the eventual control of DENV in mouse models (*Chen et al., 2013*; *Costa et al., 2017*; *Shresta et al., 2004*).

Given that the ex vivo response of pDCs was biased toward IRF7-mediated IFN-I production (*Figures 1* and *3A–B*), we hypothesized that the early ISG response in pDC:Irf7$^+$ mice was mediated by this pathway. To formally test the dependence of this early ISG response on IFN-I and thus IFNAR signaling, we employed an IFNAR1 blocking antibody and challenged mice with DENV. IFNAR blockade in pDC:Irf7$^+$ mice abrogated this early ISG response (*Figure 5—figure supplement 1A*) and correspondingly increased both DENV viremia in these mice, as readily observed in the spleen (*Figure 5—figure supplement 1C*) and to a lesser extent in the blood (*Figure 5—figure supplement 1D*; not significant; likely as the consequence of the limited dynamic range of DENV detection at this time point in blood). Of note, consistent with the results of *Figure 5*, ISG expression was still detected in the Irf3/7 DKO and pDC:Irf7$^+$ at a later time point (72 h p.i.) (*Figure 5—figure supplement 1A–B*), further suggesting that the late induction of ISGs is independent of pDC IRF7-mediated IFN-I and likely induced by IFN-II. Along the same line, recent work demonstrates that IRF1-mediated IFN-II can be induced by DENV, independent of *Irf3/Irf7* expression (*Carlin et al., 2017*). In accordance, we showed that the IFNAR1-dependent response contributes to the control of DENV in Irf3/7 DKO mice (*Figure 5—figure supplement 1C*).

Together, our results demonstrated that, despite undetectable IFN-I at a systemic level, the early ISG induction and control of DENV by the pDC-IRF7-induced response relies on IFN-I/IFNAR1 signaling. Importantly, we also observed an acceleration of IFN-II signaling as a result of IRF7 expression in pDCs, temporally associated with viral control in DENV-infected pDC:Irf7$^+$ mice.

## Mediation of IFN-II production and viral control by NK cells

We next aimed at identifying the cell type(s) responsible for IFN-II production and its subsequent impact on signaling in infected pDC:Irf7$^+$ mice. We observed that an elevated fraction (~12%) of splenic NK cells produced IFNγ in DENV-infected pDC:Irf7$^+$ mice (*Figure 6A*). For other cell types, including NKT cells, γδ-T cells, αβ-T cells and neutrophils, IFNγ+ cells represented an extremely small fraction of total splenocytes (*Figure 6A*, right panel and S5A-B). Interestingly, splenic, but not blood-circulating NK cells, produced IFNγ protein in response to DENV infection, consistent with the absence of upregulation of blood *Ifng* mRNA (data not shown) and limited induction of the activation marker CD69 on blood NK cells (*Figure 6—figure supplement 1C*). This implies that NK cell response is localized to the spleen, the primary site of DENV replication. In agreement with a localized host response, we showed that the accelerated IFN-II response in DENV-infected pDC:Irf7$^+$ mice was associated with an early recruitment of neutrophils to the spleen (*Figure 6—figure supplement 1D*) as well as maturation of monocytes (identified as Ly6C$^{++}$/Lineage$^-$ cells), as represented by upregulation of MHCII, which did not occur in Irf3/7 DKO mice (*Figure 6—figure supplement 1E*). Of note, both cell types have been previously shown to be involved in the modulation of NK cell activation (*Costa et al., 2017*; *Kang et al., 2008*).

As NK cells were the primary producers of IFN-II (*Figure 6A*), we sought to identify their role during DENV infection in pDC:Irf7$^+$ mice in NK cell depletion studies. An efficient depletion of NK cells (i.e. reduction by >80% of splenic and blood NK cells, *Figure 6B* and *Figure 6—figure supplement 2A*) lead to a significant increase of both splenic DENV RNA and viral titers in pDC:Irf7$^+$ mice

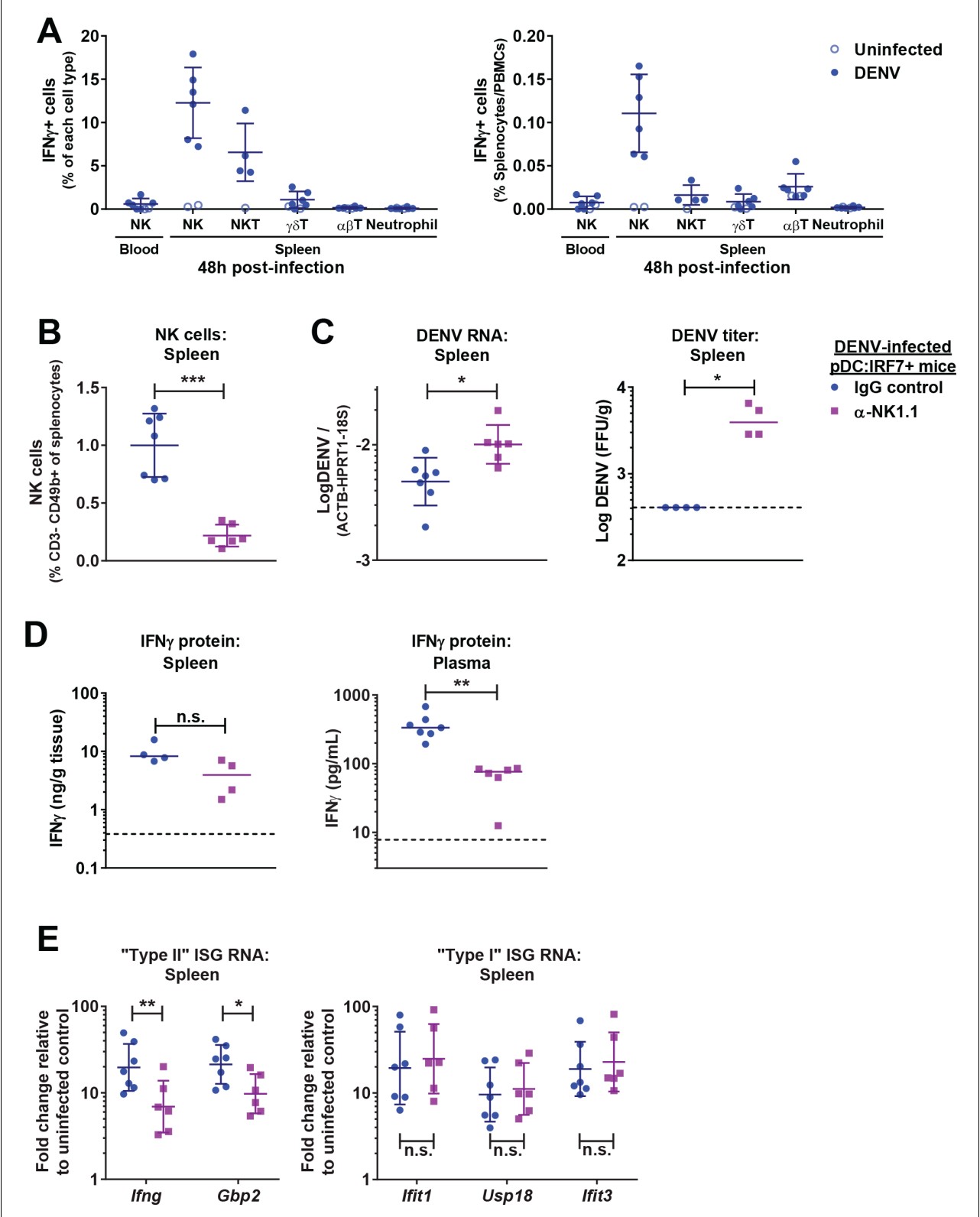

**Figure 6.** NK cells produce IFNγ and control DENV infections in pDC:Irf7[+] mice. (**A**) Quantification of IFNγ[+] cells in spleen/blood samples from DENV infected pDC:Irf7[+] mice at 48 h p.i. by FACS. Cell subsets were identified as shown in *Figure 6—figure supplement 1A*. Results presented as a percentage of each cell type (left panel) or relative to total splenocytes/PBMCs (right panel); mean ± SD, *n* = 4–6 infected, *n* = 1–2 uninfected mice per cell population. (**B–E**) pDC:Irf7[+] mice were injected i.p. with anti-NK1.1 depleting or IgG2a control antibody and infected 24 hr later with DENV i.v.;
*Figure 6 continued on next page*

*Figure 6 continued*

n = 6–7 mice per condition. Analyses were performed at 48 h p.i. in the indicated tissues. (B) NK cells as a percentage of splenocytes were identified by FACS as in **Figure 6—figure supplement 2A**; mean ± SD. (C) $Log_{10}$ DENV genome levels; mean ± SD and $Log_{10}$ foci forming unit (ffu)/g tissue ; median. (D) Quantification of IFNγ by ELISA; median. IFNγ was undetectable in uninfected control mice. (E) ISG RNA levels are expressed as fold change relative to uninfected IgG2a-treated control mice; geometric mean ± SD.
DOI: https://doi.org/10.7554/eLife.34273.012
The following figure supplements are available for figure 6:

**Figure supplement 1.** Activation of NK cells in DENV-infected mice, neutrophil influx into the spleen and monocyte activation correlate with pDC-dependent responses to DENV.
DOI: https://doi.org/10.7554/eLife.34273.013
**Figure supplement 2.** Depletion of NK cells in DENV infected mice.
DOI: https://doi.org/10.7554/eLife.34273.014

(**Figure 6C**). This was associated with a decrease in IFNγ mRNA, IFNγ protein and IFN-II ISG signature in the spleen and plasma (**Figure 6D–E**), along with reduced neutrophil recruitment to the spleen and maturation of monocytes (**Figure 6—figure supplement 2B–C**). Closing the circle and establishing IFNα responses as upstream, we observed that IFN-I-induced ISG levels were not impacted by NK cell depletion in pDC:Irf7$^+$ mice at 48 h p.i. (**Figure 6E**).

## CHIKV activates a pDC IRF7-restricted response via cell-cell contact and, which controls in vivo infection

We extended the study to CHIKV, also a positive-sense single-stranded RNA mosquito-borne virus of major concern for human health. We tested whether pDC activation by CHIKV shares the same features as DENV, showing that indeed, human pDCs produced robust IFNα in response to CHIKV only when in contact with CHIKV-infected cells (**Figure 7A**). Similarly, robust IFNα levels were detected when WT mouse pDCs were co-cultured with CHIKV infected Vero cells or with syngeneic co-cultures of Irf3/7 DKO mouse embryonic fibroblasts (MEFs) (**Figure 7B–C**). By contrast, cell-free CHIKV and CHIKV-infected cells physically separated from pDCs did not induce IFNα secretion (**Figure 7A–B**). CHIKV-infected cells failed to activate pDCs deficient for *Tlr7*, whereas pDCs deficient for *Mavs* (i.e. the downstream signaling adaptor of RLR pathway) produced levels of IFNα comparable to WT pDCs (**Figure 7D** and data not shown). Comparable to DENV-induced responses, the sensing of CHIKV-infected cells by pDCs led to an IRF7-mediated cytokine production (IFNα), but not NF-κB-induced inflammatory cytokines (TNFα) (**Figure 7E**). Together, this demonstrated that, like DENV, pDC induction by CHIKV required the cell-to-cell sensing of infected cells by TLR7-induced signaling and led to an IRF7-restricted response.

We thus tested whether pDC IRF7-induced responses were sufficient to protect mice from CHIKV. Mice were infected s.c. with CHIKV, and as previously reported WT mice showed no clinical symptoms, whereas Irf3/7 DKO mice rapidly succumbed to CHIKV infection (**Figure 7F**) (**Schilte et al., 2012**). By contrast, pDC:Irf7$^+$ mice experienced no overt clinical symptoms with 100% of mice surviving infection (**Figure 7F**). Early control of viremia was improved in pDC:Irf7$^+$ mice compared to Irf3/7 DKO mice, although reduced as compared to WT mice (**Figure 7G**). By day 7 p.i., pDC:Irf7$^+$ mice had cleared CHIKV infections (data not shown). Importantly, there was differential control of CHIKV in pDC:Irf7$^+$ mice depending on the tissue. The initial sites of CHIKV replication (i.e. the infection site: skin/muscle and muscle) displayed relatively higher viral titers as compared to more distal sites (i.e. the spleen and liver), where virus was barely detectable at 48 h p.i. in pDC:Irf7$^+$ mice (**Figure 7G**). It is possible that the pDC antiviral response controls infection once virus spreads systematically, as previously proposed in the context of Herpes Simplex virus infection (**Swiecki et al., 2013**).

We established a mouse IFNα Single Molecule Assay (SiMoA) (**Rissin et al., 2010**), with a quantification limit at 0.1 pg/ml (i.e. >100 fold increased sensitivity compared to current assays, data not shown). Nonetheless plasma IFNα remained undetectable in CHIKV-infected pDC:Irf7$^+$ mice at both 24 and 48 h p.i. (data not shown). Therefore, we validated the dependence on IFN-I-induced signaling for the control of CHIK by pDCs using IFNAR1 blocking antibody. IFNAR blockade in pDC:Irf7$^+$ mice induced lethality and, consistently, greatly increased CHIKV viremia (**Figure 7—figure supplement 1A–B**). By contrast with results for DENV (**Figure 5—figure supplement 1C**), IFNAR1-

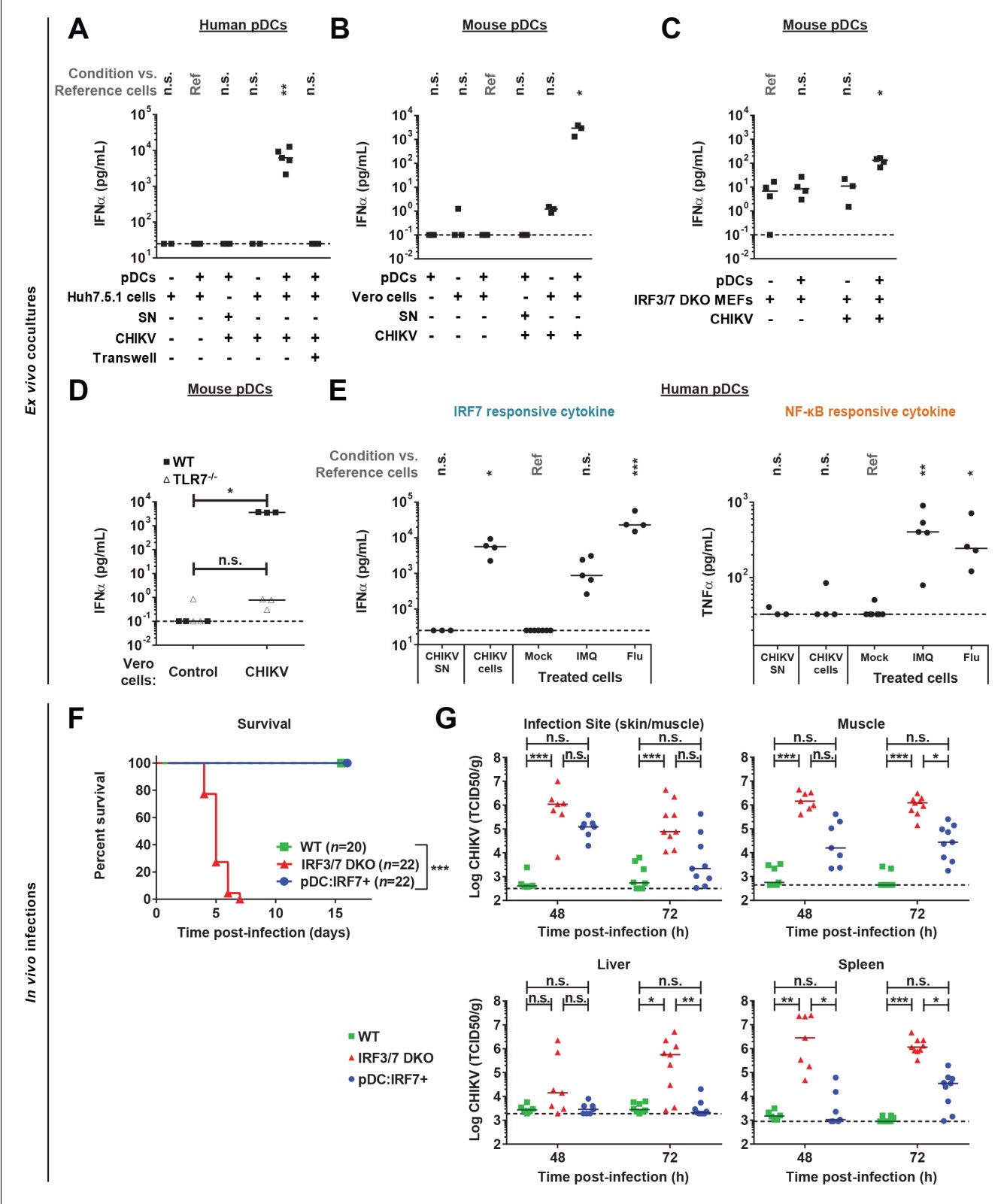

**Figure 7.** CHIKV activates a pDC IRF7-restricted response via cell-cell contact and pDC:Irf7+ mice control CHIKV viremia. (**A**) Quantification of IFNα by ELISA in supernatants of human pDCs cocultured for 22–24 hr with CHIKV-infected or uninfected Huh7.5.1 cells [seeded together or separated by a 0.4 µm transwell membrane] or treated with CHIKV infectious supernatant (SN); median, *n* = 3–5 independent experiments. (**B–D**) Quantification of IFNα levels by single molecule array (SiMoA) assay in supernatants of pDCs isolated from spleens of WT or TLR7$^{-/-}$ mice and cocultured with Vero cells (**B, D**)

*Figure 7 continued on next page*

*Figure 7 continued*

or Irf3/7 DKO MEFs (C) for 24 hr. Cultured cells were infected or not with CHIKV-GFP, as indicated; median, *n* = 3–4 independent experiments. (E) Quantification of IFNα and TNFα by ELISA in supernatants of human pDCs treated for 22–24 hr with CHIKV infectious supernatant (CHIKV SN), TLR7 agonists [IMQ; 1 μg/mL and Flu; $3 \times 10^3$ FFU], or cocultured with CHIKV-infected Huh7.5.1 cells; median, *n* = 4–5 independent experiments. (F–G) CHIKV infection subcutaneously (s.c.) of mice with the indicated genotypes followed by analysis of the survival rate (F) and CHIKV infectious titers in the indicated tissue homogenates (G); median, *n* = 7–22 mice per condition.
DOI: https://doi.org/10.7554/eLife.34273.015

The following figure supplement is available for figure 7:

**Figure supplement 1.** Protection in pDC:Irf7$^+$mice involves IFNAR1 and NK response.
DOI: https://doi.org/10.7554/eLife.34273.016

dependent response did contribute to the control of CHIKV in Irf3/7 DKO mice (*Figure 7—figure supplement 1B*). This discrepancy between DENV and CHIKV infections likely reflects the greater impact of IFN-I signaling in CHIKV versus DENV infection and/or a differential contribution of the alternate IRF-mediated signal transduction. Similar to DENV, NK cell depletion in pDC:Irf7$^+$ mice increased CHIKV titers (*Figure 7—figure supplement 1C*, ~10 fold-increase, although not statistically significant).

Together, these data indicate that IRF7 expression in pDCs is sufficient to control two genetically distant RNA viruses (CHIKV and DENV) via the local production of IFN-I and IFNγ secretion by NK cells.

## Discussion

DENV and CHIKV are important mediators of human diseases, causing a significant human and economic burden worldwide. We provide compelling evidence that pDCs are a key cell type in the initiation of antiviral responses to these two distinct RNA viruses. We demonstrate that pDC activation by DENV/CHIKV is characterized by IRF7-prioritized signaling, while passively excluding NF-κB-mediated responses. Despite undetectable levels of systemic IFN-I, pDC IRF7 signaling is critical for the induction of a potent antiviral response. This might reflect a local pDC IFN-I response and is in keeping with the requirement for physical contact with infected cells to activate pDCs, in turn inducing a potent ISG response. Notably, pDC-IRF7 activation accelerates NK-mediated IFN-II induction, which assists in control of viremia.

pDCs have previously been demonstrated to be required for the control of several viral infections, including herpes simplex virus (HSV), murine hepatitis virus (MHV), murine cytomegalovirus (MCMV), vesicular stomatitis virus (VSV) and lymphocytic choriomeningitis virus (LCMV) (*Swiecki and Colonna, 2015*). In contrast, others reported that pDCs are dispensable and/or do not contribute to IFN-I production in the case of certain infections, including influenza virus, respiratory syncytial virus (RSV), Newcastle disease virus (NDV) and MCMV (*Del Prete et al., 2015*; *GeurtsvanKessel et al., 2008*; *Jewell et al., 2007*; *Kumagai et al., 2007*; *Swiecki et al., 2013*), which may be a reflection of viral replication mechanisms, inoculum size, or location (*Swiecki et al., 2010*; *Swiecki et al., 2013*). These observations bring the emerging concept that while antiviral response (i.e. IFN-I production) in infected cells acts as a first line defense, pDCs may serve as a failsafe, triggered upon systemic infection and therefore failure of viral control (*GarciaGarcía-Sastre, 2017*; *Tomasello et al., 2014*). This assumption is in line with the multiple inhibitory mechanisms of antiviral sensing by viral proteins expressed by infected cells (*GarciaGarcía-Sastre, 2017*). Nonetheless, previous investigations have primarily demonstrated the importance of pDCs in viral control by depletion of pDCs (*Swiecki and Colonna, 2015*). Here, we validated a model consisting of a pDC-restricted IFN response, which offers a unique opportunity to define pDC function. This new model also permitted the study of the possible cryptic pDC-mediated control of certain infections. For instance, pDC antiviral function might be masked due to the homeostasis of the innate immunity, including an enhanced response by other cell types when pDCs are depleted.

We demonstrated that pDCs are sufficient to initiate early control of viral infections, via induction of IRF7-prone signaling. In this mouse model, the restoration of *Irf7* expression in pDCs does not reflect control under the endogenous promoter, as the *Irf7* gene is under the control of the pDC-specific *Siglech* promoter. However, *Irf7* overexpression artifacts are avoided in the pDC:IRF7+

model as *Irf7* steady-state expression is reduced under the control of the *Siglech* promoter relative to the *Irf7* promoter (*Figure 2B*). Furthermore, *Siglech* expression by pDCs is inversely correlated to ISG upregulation in vivo (*Figure 4—figure supplement 1E*), in agreement with prior study (*Puttur et al., 2013*). Importantly, while systemic IFN-I was not detectable in response to DENV/CHIKV infection, the pDC-IRF7 response rapidly and potently induces a broad array of ISGs in multiple organs, dependent on IFN-I signaling. Therefore, in line with our ex vivo results in this and previous studies (*Décembre et al., 2014*), we propose that pDCs respond in a contact-dependent manner to infected cells, and thus induce highly localized IFN-dependent responses. Importantly, this response is sufficient to impart a large-scale antiviral response, which protects from viral infections.

The engagement of TLR7/9 leading to IFN-I production and NF-κB-dependent pro-inflammatory cytokines by pDCs may affect downstream immunological parameters, such as for example, B cell differentiation induced by pDC response to influenza virus (*Jego et al., 2003*). We observed that pDCs fail to respond via NF-κB-mediated inflammatory responses to DENV/CHIKV-infected cells, indicating that pDC TLR7 signaling induced by these viruses primarily leads to IRF7 induction. Consistently, on the whole-organism level, while ectopic IRF7 expression in pDCs fully restores ISG responses, NF-κB responses to DENV were absent in both pDC:Irf7+ and Irf3/7 DKO mice, thus underlining the importance of the IRF7 and dispensability of the NF-κB responses in pDC-mediated control of DENV and CHIKV infections. Our results also imply that NF-κB induction by these viruses in other cell types likely requires cross-talk between IRF3 and NF-κB (*Freaney et al., 2013*; *Iwanaszko and Kimmel, 2015*).

While pDC IFN-I responses were important in early control of both DENV and CHIKV infections in vivo, substantial differences were observed in the overall outcome of infections. CHIKV infections were invariably lethal in the Irf3/7 DKO genetic background, and pDC IFN-I responses provided 100% protection to this lethal phenotype. By contrast, DENV infections were well-controlled even in Irf3/7 DKO mice by 72 h p.i. Notably, pDC IFN-I responses lead to substantial control of DENV viremia at time points intermediate between WT and Irf3/7 DKO mice (~42–48 hr). In our study, the lethal phenotype in CHIKV infection in Irf3/7 DKO mice correlated with substantially higher viral titers, as compared to DENV levels. Interestingly, blockade of IFN-I signaling in Irf3/7 DKO mice only increased viremia in situations where lower basal viral titers were observed: for example, DENV infections in all organs, or CHIKV viral titers in the liver. We thus suggest that the persistence of IFN-I-mediated viral control under these conditions depends on IRF5 or IRF1 signaling (*Carlin et al., 2017*; *Chen et al., 2013*; *Lazear et al., 2013*; *Rudd et al., 2012*), as control of flaviviral infections is still contingent on IFNAR signaling in the absence of *Irf3/Irf7* expression. The contribution of these alternate IRF proteins to the antiviral response may only be revealed in case of relatively low viremia.

Eventual control of DENV in the Irf3/7 DKO background depended on the induced IFNγ response (*Chen et al., 2013*). In agreement, we observe an IFN-II response in DENV-infected Irf3/7 DKO mice peaking at ~42 h p.i., which is largely absent in WT mice. Importantly, the antiviral IFN-II response is accelerated in pDC:Irf7+ mice relative to Irf3/7 DKO mice. These pDC-regulated IFN-II responses are consistent with prior work showing a defect in type II IFN activation in a pDC-deficient mouse model (*Guillerey et al., 2012*). We demonstrated that IFNγ production in DENV-infected pDC:Irf7+ mice is largely mediated by NK cells in the spleen, the primary site of DENV replication. These activated NK cells mediate downstream IFN-II responses that contribute to control of DENV viremia, in accordance with a recent report (*Costa et al., 2017*). A similar decrease in viremia was observed in CHIKV-infected pDC:Irf7+ mice depleted of NK cells, although these results did not reach statistical significance. DENV may be more sensitive to type II IFN signaling (relative to type I IFN) compared to CHIKV, as DENV infections cause higher lethality in *Ifngr-/-* compared to *Ifnar-/-* mice, a situation that is reversed in the context of CHIKV infection (*Gardner et al., 2012*; *Shresta et al., 2004*).

Importantly, IFNγ production by NK cells was not observed in the blood, a site of low DENV viremia, likely indicating a necessity for NK cells to be closely apposed to DENV-infected cells and/or DENV-activated cells to achieve NK cell activation of IFN-II responses. We also observed an association with other immune cell parameters, including a pDC IRF7-mediated influx of neutrophils and monocyte activation in the spleens of DENV-infected mice. While some reports suggested that type I IFN directly contributes to NK cell response, NF-κB-dependent cytokines are well-known to regulate IFNγ production by NK cells (*Schulthess et al., 2012*). Along the same lines, in vitro studies suggested that pDCs act in concert with monocytes to potentiate NK-cell-mediated IFNγ production in

the context of HCV and DENV infection (*Costa et al., 2017*; *Zhang et al., 2013*). Future studies are required to determine how DENV/CHIKV-activated pDCs can directly augment NK cell IFNγ responses, or whether pDC-produced type I IFN acts through other cell types and attendant NF-κB-dependent cytokine responses for this amplification of type II induction.

The ability to segregate NF-κB and IRF7 responses may have potential use in developing novel therapies. Notably, NF-κB-dependent cytokines have been associated with disease severity in both DENV and CHIKV infection (*Chen et al., 2007*; *Dupuis-Maguiraga et al., 2012*). This is juxtaposed with pDC-derived IFN responses supporting viral clearance while avoiding possible inflammatory responses. As arboviral infections tend to be acute, rather than chronic diseases, this study provides an important indication of how early viral control may be mediated by one of the key drivers of the IFN response, the plasmacytoid dendritic cell.

## Materials and methods

### Cells and reagents

Huh7.5.1 cells (*Zhong et al., 2005*) (RRID:CVCL_E049), Vero cells (CLS Cat# 605372/p622_VERO, RRID:CVCL_0059), and BHK-21 cells (ATCC Cat# CCL-10, RRID:CVCL_1915) were maintained in DMEM (Dulbecco's modified Eagle medium; Life Technologies) supplemented with 10% FBS, 100 units (U)/ml penicillin, 100 mg/ml streptomycin, 2 mM L-glutamine (Life Technologies). Cells were maintained at 37°C in 5% $CO_2$. MEFs were maintained in supplemented RPMI growth media. The work involved cell line, with routinely verify status for contamination (e.g. Mycoplasma).

Cell culture reagents included the TLR7 agonist (IMQ; imiquimod) and TLR7/8 agonist (R848) (Invivogen, San Diego, CA); TLR7 antagonist, IRS661 (5'-TGCTTGCAAGCTTGCAAGCA-3') synthesized on a phosphorothioate backbone (MWG Biotech, Ebersberg, Germany); TLR3 agonist (poly-I:C) (Invivogen); TLR9 agonist (CpG) (Invivogen); recombinant IFNβ (PBL Interferon Source, Piscataway, NJ). General reagents included: paraformaldehyde (PFA) and brefeldin A (BFA) (Sigma-Aldrich, St Louis, MO); Ficoll-Hypaque (GE Healthcare Life Sciences, Piscataway, NJ).

### Viral stocks

Viral stocks of the prototypic DENV-2 strain New Guinea C (NGC) (AF038403) were produced using in vitro RNA transcripts prepared from DENV-2 infectious plasmid clone pDVWS601 plasmid (*Pryor et al., 2001*). Briefly, plasmid was linearized with XbaI (New England Biolabs, Ipswich, MA) and RNA transcripts were produced and purified using mMESSAGE mMACHINE T7 Kit (Ambion/Thermo Fisher Scientific, Waltham, MA). RNA transcripts were introduced into BHK-21 cells by electroporation (using 100 μg RNA to transfect $8 \times 10^7$ cells, with $4 \times 10^6$ cells per electroporation). Electroporation was performed in 0.4 cm cuvettes (Bio-Rad, Hercules, CA) with a Gene Pulser system (Bio-rad), using a square-wave pulse (280 V, 25 ms). Virus-containing supernatants were collected at 3–5 days post-electroporation and clarified through a 0.2 μm filter (Corning Inc, Corning, NY). Viral supernatants were concentrated by ultracentrifugation at 75000 x *g* for 2 hr at 4°C, resuspended in DMEM growth media, and frozen at −80°C until titration and/or use. Viral stocks were titered on Huh7.5.1 cells, as described below, and adjusted to $2 \times 10^7$ focus-forming units (ffu)/mL.

CHIKV-21 viral stocks, used for all in vivo infections, were prepared from clinical samples (*Schilte et al., 2010*). Briefly, CHIKV-21 was propagated in C6/36 mosquito cells, and infectious supernatants were harvested, clarified through a 0.2 μm filter, and frozen at −80°C until titration and/or use. Recombinant CHIKV-GFP, which expresses GFP under a subgenomic promoter (CHIKV-GFP 5'), was generated from a full-length infectious cDNA clone provided by S. Higgs (*Vanlandingham et al., 2005*) and was used for all CHIKV in vitro mouse studies. Briefly, CHIKV-GFP 5' plasmid was linearized with *NotI* (New England Biolabs) and RNA transcripts were produced and purified using mMESSAGE mMACHINE SP6 Kit (Ambion). RNA transcripts were introduced into BHK-21 cells by electroporation (using 1 μg RNA to transfect $5 \times 10^6$ cells). Virus-containing supernatants were collected when the viral cytopathic effect reached 75% of transfected cells, clarified through a 0.2 μm filter (Corning Inc), and frozen at −80°C until titration and/or use. Viral stocks were titered on Vero cells, as described below.

DENV infectivity titers in concentrated culture supernatants or mouse organ homogenates were determined by end-point dilution using Huh 7.5.1 cells (*Décembre et al., 2014*). Foci forming units

(ffu) were detected 72 hr after infection. Briefly, Huh7.5.1 cells were fixed with 4% PFA and permeabilized by incubation for 7 min in PBS containing 0.1% Triton. Cells were then blocked in PBS containing 3% BSA for 15 min and incubated for 1 hr with mouse anti-E glycoprotein (clone 3H5) hybridoma supernatant diluted in PBS containing 1% BSA. After three washes with PBS, cells were incubated 1 hr with secondary Alexa 555-conjugated anti-mouse antibody. Virally-infected foci of E-positive cells were determined using a Zeiss Axiovert 135 microscope.

CHIKV infectivity titers in culture supernatants or mouse organ homogenates were titrated as $TCID_{50}$ endpoint on Vero cells using a standard procedure. Serial 10-fold dilutions of supernatants were added in six replicates in 96-well plates seeded with $10^4$ Vero cells. The cytopathic effect was scored 5 days after infection and the titers were calculated by determining the last dilution giving 50% of wells with cells displaying a cytopathic effect. Results were expressed as $TCID_{50}$/mL.

Viral stocks of Influenza A Virus (FluAV, H1N1/New Caledonia/2006) were produced as previously described (*de Chassey et al., 2013*) and kindly provided by Dr V. Lotteau (CIRI, Lyon, France).

## Mouse models, infection and treatment

All mice, previously backcrossed on a C57BL/6 background, were bred and maintained at the Institut Pasteur (Paris, France) and the Plateau de Biologie Expérimentale de la Souris (PBES, Lyon, France) with Authorization Agreement C 69 123 0303. All animal studies were performed in accordance with the European Union guidelines for approval of the protocols by the Institutional Committees on Animal Welfare of the Institut Pasteur (Paris, France, OLAW assurance #A5476-01) and by the local ethics committee Rhône-Alpes d'Ethique pour l'Expérimentation Animale (Lyon, France, Authorization Agreement C2EA-15).

Housing conditions for mice in the PBES were as follows. Mice were housed under group conditions in individually ventilated cages, from 1 to 8 single-sex mice per cage. Specific-pathogen free (SPF) housing was maintained, with regular sanitary status testing of sentinel mice and bedding for standard mouse pathogens. Mice were fed LASQCdiet Rod16-R (Lasvendi), and temperatures were maintained at 22 ± 2°C. All mice were treatment- and drug-naive at the time of experimental manipulation. Manipulations including and following viral infections were carried out under animal BSL-3 (A3) conditions; mice were anesthetized with xylazine/ketamine (20 mg/kg and 100 mg/kg by body weight) for all A3 manipulations. Similar housing conditions were maintained at the Institut Pasteur: SPF housing, 1–5 single-sex mice per cage, and manipulations carried out under animal BSL3. All in vivo CHIKV experiments were performed at the Institut Pasteur.

*Irf3^{-/-}Irf7^{-/-}* double knockout (referred to as Irf3/7 DKO) mice were generated and kindly provided by Dr T. Taniguchi (University of Tokyo, Tokyo, Japan) (*Honda et al., 2005*). *cardif-/-* mice were generated by Dr J. Tschopp, and *Tlr7^{-/-}* mice were provided by Dr S. Akira (Osaka University, Osaka, Japan). Knock-in *Siglech^{Irf7+}* mice were generated by Ozgene Pty Ltd company. Next, knock-in *Siglech^{Irf7+}* mice were backcrossed onto the Irf3/7 DKO background strain for approximately 10 generations to generate *Siglech^{Irf7+}*Irf3/7 DKO breeding stocks, referred to as 'Union Irf3/7 DKO'. As the *Siglech^{Irf7+}* transgene replaced the endogenous *Siglech* gene, to preserve *Siglech* expression and functionality in experimental mice, 'Union Irf3/7 DKO' mice were bred with Irf3/7 DKO mice to generate *Siglech^{Irf7/+}* Irf3/7 DKO ('pDC:Irf7^{+}') F1 mice.

For in vivo synthetic agonist experiments, 100 μg of CpG 2216 (Invivogen, tlrl-hodna)+DOTAP (Roche) or 10 μg of poly-I:C (Invivogen, tlrl-pic) were injected intravenously (i.v.) into 8- to 12-week-old mice. For DENV in vivo experiments, 8- to 16-week-old mice were infected systemically with DENV. 1–3 litters of mice of each genotype were used per experiment; each litter was aged within 3 weeks of all others in the same experiment. Mice were randomly assorted into treatment groups by the experimenter. $2 \times 10^6$ ffu of DENV was injected i.v. in the retro-orbital sinus using a 31-gauge 0.3 mL insulin syringe (Becton-Dickinson, Franklin Lakes, NJ) for all experiments. In *Figure 3—figure supplement 1*, an additional group of mice were i.v. infected with $2 \times 10^5$ ffu DENV, as indicated. DMEM growth media was used as a negative injection control. For CHIKV experiments, 8- to 12-week-old adult mice were infected with $10^6$ pfu (plaque-forming units) of CHIKV-21 sub-cutaneously in the right flank.

For IFNAR blocking studies, a single dose (800 μg/mouse) of anti-IFNAR1 antibody (Leinco Technologies, Fenton, MO; clone MAR1-5A3) or $IgG_1$ isotype control was administered i.p. 24 hr (CHIKV) or 3 hr (DENV) before infection.

For NK cell depletion, a single dose (800 µg and 200 µg for CHIKV and DENV experiments, respectively) of α-NK1.1 antibody (BioXCell, West Lebanon, NH; clone PK136) or IgG$_{2A}$ isotype control was injected i.p. 24 hr before infection. Mice cohorts, representing individual experiments, are provided in *Supplementary file 1*, with mice age and sex.

## Processing and analysis of mouse tissues

Solid tissues were collected from mice and cell suspensions were prepared by pressing tissues through 40 µm cell strainers (Becton-Dickinson) or homogenates were prepared with the TissueLyser II (Qiagen, Hilden, Germany) or GentleMACs (Miltenyi, Bergisch-Gladbach, Germany) systems. Blood was collected by cardiac puncture (end-point) or retro-orbital bleeds and anticoagulated with ethylaminediamine tetra-acetic acid (EDTA) prior to centrifugation to collect formed elements ('blood') or plasma. For viral titers and protein (ELISA) analysis, tissues were homogenized in PBS, and for qRT-PCR analysis, tissues were homogenized in guanidinium thiocyanate citrate buffer (GTC) (*Dreux et al., 2012*). Virus from tissue homogenates was titrated on Vero cells (CHIKV) or Huh7.5.1 cells (DENV). Results are expressed as median culture infective dose TCID50/g (CHIKV) or ffu/g (DENV).

## Cytokine measurements

IFNα/β activity from mouse tissue homogenates or plasma was quantified using Luciferase assay (Promega, Madison, WI). Briefly, type I IFN levels were determined by incubating the reporter cell line LL171 with diluted plasma or tissue homogenates (in RPMI 1640 medium supplemented with 10% FCS, 2 mM L-glutamine, 50 mM β-mercaptoethanol) for 8 hr. Type I IFN activity was calculated using serial dilutions of a recombinant standard (IFNα4; PBL Interferon Source, Piscataway, NJ). Cytokine protein measurements from tissue culture supernatants, plasma, and tissue homogenates were performed by ELISA using commercially available kits: human IFNα and mouse IFNα/IFNβ (PBL Interferon Source), human TNFα and mouse TNFα/IL6/IFNγ (eBioscience/Thermo Fisher Scientific, Waltham, MA). When indicated, mouse IFNα protein levels were determined using a newly developed mouse IFNα assay on the Quanterix SiMoA platform using capture and detection antibodies obtained from eBioscience (capture: Mouse IFNα Platinum ELISA capture antibody; detection: Mouse IFNα Platinum ELISA detection antibody). Recombinant mouse IFNα2 (eBioscience) was used as a standard.

## Quantitative reverse transcription-PCR (qRT-PCR) analysis

RNAs were isolated from plasma or tissue homogenates prepared in guanidinium thiocyanate citrate buffer (GTC; Sigma-Aldrich) by phenol/chloroform extraction procedure as previously (*Dreux et al., 2012*). Reverse transcription was performed using the random hexamer-primed High-Capacity cDNA reverse transcription kit (Applied Biosystems, Foster City, CA) and quantitative PCR was carried out using the Powerup Sybr Green Master Mix (Applied Biosystems). For DENV and ISG analysis in mouse spleen, liver, and blood, absolute numbers of transcripts were generally normalized to the geometric mean of *β-actin, hprt1,* and *18S* housekeeping gene transcript numbers. For plasma samples lacking housekeeping transcripts, qRT-PCR was controlled by the addition of xenogeneic carrier RNAs encoding *xef1α* (xenopus transcription factor 1α) in vitro transcripts in plasma diluted in GTC buffer. The sequences of the primers used in analysis are described in *Supplementary file 2*.

## Isolation of pDCs and ex vivo coculture experiments

Mouse pDCs were isolated from male and female mice from 8 to 22 weeks of age, of the indicated genotypes. pDCs were isolated from splenocytes/bone marrow via negative selection using the pDC isolation kit II (Miltenyi) or from bone marrow via positive selection using PDCA1-biotin antibody (Miltenyi) followed by anti-biotin microbead (Miltenyi) selection. Human pDCs were isolated from PBMCs derived from leukapheresis of healthy adult human volunteers, obtained according to procedures approved by the 'Etablissement Français du sang' (EFS) Committee. PBMCs were isolated using Ficoll-Hypaque density centrifugation. Human pDCs were positively selected from PBMCs using BDCA-4-magnetic beads (Miltenyi), as we previously reported (*Décembre et al., 2014*).

After isolation, pDCs were cultured with Vero, Huh7.5.1, BHK-21, or Irf3/7 DKO primary MEF cells in RPMI cell growth media (10% FBS, 100 U/ml penicillin, 100 mg/ml streptomycin, 2 mM

L-glutamine, non-essential amino acids, 1 mM sodium pyruvate and 0.05 mM β-mercaptoethanol) in 96-well round bottom plates at 37°C, as previously described (*Décembre et al., 2014*), using $8 \times 10^4$ mouse pDCs when cocultured or $1 \times 10^4$ human pDCs. Cocultured cells were either naive, previously infected or infected 24 hr post-coculture, using a multiplicity of infection of 5 and 2 for CHIKV and DENV, respectively. Cell culture supernatants were collected at 22–24 hr after the beginning of coculture with the infected cells.

## Antibodies, flow cytometry and cell sorting

For flow cytometry, FACs sorting, and magnetic cell isolation, biotin, BV421, Pacific Blue, FITC, PE, PE-Cy7, APC, AlexaFluor-647, AlexaFluor-700, APC-eFluor780 or PerCP-Cy5.5 conjugates of the following anti-mouse antibodies were used (clone in parentheses): Siglec-H (440 c), B220 (RA3-6B2), CD11c (N418), CD11b (M1/70), CD8a (53–6.7), CD2 (RM2-5), CD19 (1D3), CD4 (GK1.5), CD3 (17A2, 500A2), NK1.1 (PK136), CD69 (H1.2F3), Ly6C (HK1.4), Ly6G (1A8), PDCA-1 (JF05-1C2.4.1), CD25 (3C7), CD49b (DX5, HMα2), TER-119 (TER-119), TCR-γδ (UC7-13D5), TCR-β (H57-597), IFNγ (XMG1.2), IRF7 (MNGPKL) (Miltenyi; eBioscience; Becton-Dickinson; Biolegend, San Diego, CA). Dylight-405 and Dylight-680 streptavidin conjugates (Life Technologies) were used to detect biotinylated antibodies.

Intracellular and extracellular staining was performed as follows. Isolated splenocytes and blood cells were depleted for erythrocytes by incubation at room temperature in red blood cell lysis buffer (1.5 M $NH_4Cl$, 100 mM $NaHCO_3$, 10 mM $Na_2EDTA$, pH 7.4) and cells were washed in FACS buffer (PBS + 2 mM EDTA +5% FCS). For extracellular staining, cell surface marker staining was performed with antibodies diluted in FACS buffer, followed by fixation using 4% PFA. For intracellular staining, cells were incubated for 3 hr at 37°C in RPMI cell growth media + 10 µg/mL BFA, and cell surface marker staining was performed in FACS Buffer +10 µg/mL BFA. Fixation, permeabilization, and intracellular staining (IFNγ) was performed using the Cytofix/Cytoperm kit (Becton-Dickinson).

## Statistical analysis

Statistical analysis was performed using PRISM v7.03 software (Graphpad, La Jolla, CA); only biological replicates representing separate mice and/or human donors are presented as data herein. Technical replicates, representing repeated measurements or treatments of the same cellular populations, were averaged prior to analysis. Mouse experiments were designed to detect a two-fold difference in means of parametric data at 95% confidence ($n \geq 4$ per condition, assumed CV = 0.3); however, larger sample sizes were used as litter numbers permitted. Viral titers and cytokine levels were considered non-parametric due to the presence of many data points at the assay detection limits; therefore, non-parametric Kruskal-Wallis tests with Dunn's multiple comparison corrections were performed on this data. Viral RNA levels and transcript fold changes, as log-normal parametric data, were log-transformed prior to analysis using parametric tests. Kinetic analyses of ex vivo pDC cytokines (*Figure 1A–B*) were analyzed by one-way ANOVA parametric tests of AUC (area under the curve). Parametric tests were: unpaired Student's t test for comparisons of two conditions, one- or two-way ANOVA with a Holm-Sidak multiple comparison correction for comparison of multiple conditions. Statistical analysis of survival curves was performed using log-rank (Mantel-Cox) tests.

## Acknowledgements

We thank A Davidson (Bristol University) for providing the DENV-2 strain New Guinea C, P Despres (Pasteur Institut, Paris, France) for the anti-E DENV 3H5 antibody, V Lotteau (CIRI, Lyon, France) for Influenza A virus stocks, FV Chisari (Scripps Research Institute, La Jolla, CA) for the Huh-7.5.1 cells, T Taniguchi (University of Tokyo, Tokyo, Japan) for the Irf3/7 DKO mice, J Tschopp for the *cardif*$^{-/-}$ mice, and S Akira for the *Tlr7*$^{-/-}$ mice. We are grateful to Y Jaillais for critical reading of the manuscript, F Fusil for assistance with mice and to our colleagues for encouragement and help. We acknowledge the contribution of SFR Biosciences (UMS3444/CNRS, US8/Inserm, ENS de Lyon, UCBL) facilities, PBES and cytometry for mice housing and technical assistance. This work was supported by grants from the 'Agence Nationale pour la Recherche' (ANR-JCJC-EXAMIN), the ''Agence Nationale pour la Recherche contre le SIDA et les Hépatites Virales'' (ANRS-AO 2016–01) and the LabEx Ecofect Grant ANR-11-LABX-0048 to MD, ANR-14-CE14-0015-02 CHIKV-Viro-Immuno to

MLA. BW's postdoctoral fellowship was sponsored by EMBO (ALTF 1466–2014). The funders had no role in study design, data collection and analysis, decision to publish, or preparation of the manuscript.

## Additional information

### Funding

| Funder | Grant reference number | Author |
| --- | --- | --- |
| Agence Nationale de la Recherche | ANR-JCJC-EXAMIN | Marlène Dreux |
| Agence Nationale de Recherches sur le Sida et les Hepatites Virales | ANRS-AO 2016-01 | Marlène Dreux |
| Agence Nationale de la Recherche | ANR-11-LABX-0048 | Marlène Dreux |
| Agence Nationale de la Recherche | ANR-14-CE14-0015-02 CHIKV-Viro-Immuno | Matthew L Albert |

The funders had no role in study design, data collection and interpretation, or the decision to submit the work for publication.

### Author contributions

Brian Webster, Conceptualization, Formal analysis, Validation, Investigation, Visualization, Methodology, Writing—original draft, Writing—review and editing; Scott W Werneke, Conceptualization, Formal analysis, Validation, Investigation, Visualization, Methodology, Writing—review and editing; Biljana Zafirova, Conceptualization, Investigation, Methodology; Sébastien This, Séverin Coléon, Elodie Décembre, Investigation, Methodology; Helena Paidassi, Formal analysis, Supervision, Writing—review and editing; Isabelle Bouvier, Darragh Duffy, Formal analysis, Supervision; Pierre-Emmanuel Joubert, Formal analysis, Methodology; Thierry Walzer, Resources, Supervision, Writing—review and editing; Matthew L Albert, Conceptualization, Supervision, Funding acquisition, Writing—review and editing; Marlène Dreux, Conceptualization, Supervision, Funding acquisition, Visualization, Writing—original draft, Writing—review and editing

### Author ORCIDs

Brian Webster http://orcid.org/0000-0002-6806-0363
Helena Paidassi http://orcid.org/0000-0001-9915-4365
Thierry Walzer http://orcid.org/0000-0002-0857-8179
Marlène Dreux https://orcid.org/0000-0002-6607-4796

### Ethics

Animal experimentation: All mice were bred and maintained at the Institut Pasteur (Paris, France) and the Plateau de Biologie Expérimentale de la Souris (PBES, Lyon, France) with Authorization Agreement C 69 123 0303. All animal studies were performed in accordance with the European Union guidelines for approval of the protocols by the Institutional Committees on Animal Welfare of the Institut Pasteur (Paris, France, OLAW assurance #A5476-01) and by the local ethics committee Rhône-Alpes d'Ethique pour l'Expérimentation Animale (Lyon, France, Authorization Agreement C2EA-15). Housing conditions for mice in the PBES were as follows. Mice were housed under group conditions in individually-ventilated cages, from 1-8 single-sex mice per cage. Specific-pathogen free (SPF) housing was maintained, with regular sanitary status testing of sentinel mice and bedding for standard mouse pathogens. Mice were fed LASQCdiet® Rod16-R (Lasvendi). All mice were treatment- and drug-naïve at the time of experimental manipulation. Manipulations including and following viral infections were carried out under animal BSL-3 (A3) conditions; mice were anesthetized with xylazine/ketamine (20 mg/kg and 100 mg/kg by body weight) for all A3 manipulations. Similar housing conditions were maintained at the Institut Pasteur: SPF housing, 1-5 single-sex mice per cage,

and manipulations carried out under animal BSL3. All in vivo CHIKV experiments were performed at the Institut Pasteur.

## Decision letter and Author response

Decision letter https://doi.org/10.7554/eLife.34273.021
Author response https://doi.org/10.7554/eLife.34273.022

## Additional files

### Supplementary files

• Supplementary file 1. Mouse cohorts used for *in vivo* studies. Mouse genotype, age, sex, and number are provided for the *in vivo* experiments performed in *Figures 3–7* and supplements.
DOI: https://doi.org/10.7554/eLife.34273.017

• Supplementary file 2. Primer sets used in quantitative RT-PCR. The primer sets used to amplify genes from cDNA for expression analysis are provided.
DOI: https://doi.org/10.7554/eLife.34273.018

### Data availability

All data generated or analysed during this study are included in the manuscript and supporting files.

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
