## [Decision Letter]

Thank you for submitting your article "Plasmacytoid Dendritic Cells Control Dengue and Chikungunya Virus Infections via IRF7-regulated interferon responses" for consideration by *eLife*. Your article has been reviewed by two peer reviewers, and the evaluation has been overseen by a Reviewing Editor and Wenhui Li as the Senior Editor. The following individual involved in review of your submission has agreed to reveal his identity: Jonathan J Miner (Reviewer #1).

The reviewers have discussed the reviews with one another and the Reviewing Editor has drafted this decision to help you prepare a revised submission.

This manuscript was reviewed by two experts in the field and a member of the Board of Reviewing Editors (BRE). All reviewers felt the manuscript has the potential to be an important contribution to the literature. However, the reviewers had several comments that need to be addressed before the paper can be considered further for publication in *eLife*. While the complete, unedited reviews are given below, the reviewers and the BRE member have come up with a consensus statement that outlines the priority issues that need to be addressed in a revised manuscript.

1) While describing experiments with two viruses is a strength of the manuscript, the manuscript would be improved by focusing on one virus throughout the manuscript. Experiments addressing key points could be shown for the other virus but the way that it is presented, going back and forth between the two viruses throughout the paper is confusing and distracting, especially since there are different gene expression signatures between the viruses. This approach would help in condensing and focusing the Introduction as well.

2) Some of the experiments appear to lack sufficient data points to allow for appropriate statistical analysis, calling into question the validity of the interpretations.

In a revised manuscript, the authors need to address all comments of the reviewers as stated in their complete reviews as indicated below.

*Reviewer #1:*

This is a very interesting manuscript, and I think it would be an outstanding addition to the literature. The authors demonstrate that IRF7 expression in pDCs is sufficient to restrict CHIKV and DENV infection in mice. I think the most important contribution of this paper is the discovery that an IRF7-mediated anti-viral response in pDCs is sufficient to restrict CHIKV and DENV infection in vivo.

I do not have any problem with the data, but I do have two major questions that center on methods and the Discussion.

1) In SiglecH IRF7/IRF7 mice, expression of IRF7 is not controlled by the IRF7 promoter. SiglecH promoter-controlled expression of IRF7 in pDCs may have very distinct effects from regulated expression of IRF7 by the IRF7 promoter. The manuscript would be strengthened if the authors could discuss limitations associated with this particular issue.

2) If I understood the methods correctly, mice of different genotypes appear to have been infected at an age range of up to 3 weeks. This difference in ages can make a substantial impact on vulnerability to viral infections (for example, during West Nile virus infection, infection at age 5 weeks results in ~90% lethality, whereas infection at 8 weeks results in ~75% lethality.) Instead of providing a range of ages, could the authors be more precise about the age of infected mice of each genotype?

*Reviewer #2:*

The manuscript by Webster et al. describes the assessment of pDCs in two different arboviruses: dengue (DENV) and chikungunya (CHIKV) through the role of IRF7-driven IFN production. In general, the study is technically well-conducted and to decipher the immunomodulatory functions of pDCs in arbovirus sensing, authors generated the pDC:Irf7^+^ mouse model in C57BL/6 mice in order to restrict the role of IRF7 to pDCs. A wealth of data was produced that showed an indirect role for pDC IRF7-signalling in both viruses. However, the authors should consider removing one virus to streamline the datasets to improve clarity. As it stand, the datasets are scattered and all over the place due to the different viral characteristics of the 2 viruses. Specific comments below:

1) Introduction – although well-written, the Introduction is too long and clumsy. This is expected as due to the description of 2 viruses. Both are different with specific characteristics. It could be significantly shortened and more focused.

2) Figures 1 and 2 on cell-cell sensing of DENV and CHIKV – the infection conditions and viral kinetics between these 2 viruses are very different. Where infections conducted using the same batch of cells? Nonetheless, have the authors done a direct comparison between DENV and CHIKV in terms of fold change in IFNa levels? At 24h, 1 out of 3 replicates had TNFa levels of closed to 100 pg/ml. Is this sufficient to conclude that there was minimal or no TNFa production? In general, there are insufficient replicates with no statistics done.

3) Figures 3 to 7 on in vivo responses of DENV and CHIKV in pDC:Irf7^+^ mice – a mix of DENV and CHIKV datasets are distributed between the main and supplementary figures. Insufficient data points for proper statistics to be performed that weakened the study. Have the authors considered putting the levels of ISGs from 2 different types of infections together as a comparative? This will be more insightful to understand the similar and different mechanistic effects of virus sensing on IRF signalling on pDCs. Heat-maps provided in Figure 4 and Figure 3—figure supplement 1 suggest different signatures in terms of fold change. Otherwise, the rationale of combining 2 viruses into 1 story is not compelling.

---

## [Author Response]

This manuscript was reviewed by two experts in the field and a member of the Board of Reviewing Editors (BRE). All reviewers felt the manuscript has the potential to be an important contribution to the literature. However, the reviewers had several comments that need to be addressed before the paper can be considered further for publication in eLife. While the complete, unedited reviews are given below, the reviewers and the BRE member have come up with a consensus statement that outlines the priority issues that need to be addressed in a revised manuscript.1) While describing experiments with two viruses is a strength of the manuscript, the manuscript would be improved by focusing on one virus throughout the manuscript. Experiments addressing key points could be shown for the other virus but the way that it is presented, going back and forth between the two viruses throughout the paper is confusing and distracting, especially since there are different gene expression signatures between the viruses. This approach would help in condensing and focusing the Introduction as well.

We thank the Editor for highlighting that comparing the two viruses is a strength of the study. To improve the flow of the manuscript, the text and figures were reorganized to reflect the primary focus of the manuscript on dengue virus and a corroborating figure (New Figure 7) presenting the extension of these findings in the context of chikungunya virus.

Here is the list of the changes made in the set of Figures:

- New Figure 1: previous Figure 2B-E (DENV) and the next figures are shifted as follows: new Figure 2 include results of previous Figure 3; new Figure 3=previous 4; new Figure 4=previous 5, etc.

- New Figure 7 (CHIKV data): including results of previous Figures 1A, 1B, 2A, 5A and 5B.

- New Figure—figure supplement 1 (CHKV data): including results of previous Figures 1C, 1D, S4E, S4F, 7G.

The text has been edited accordingly, with a condensed Introduction.

2) Some of the experiments appear to lack sufficient data points to allow for appropriate statistical analysis, calling into question the validity of the interpretations.

This point was addressed by performing additional experiments and statistical analyses for the results of new Figure 1A-B, Figure 2B-C, Figure 3A-B, Figure 6B-E, Figure 6—figure supplement 2 and Figure 7A-E.

In a revised manuscript, the authors need to address all comments of the reviewers as stated in their complete reviews as indicated below.Reviewer #1:This is a very interesting manuscript, and I think it would be an outstanding addition to the literature. The authors demonstrate that IRF7 expression in pDCs is sufficient to restrict CHIKV and DENV infection in mice. I think the most important contribution of this paper is the discovery that an IRF7-mediated anti-viral response in pDCs is sufficient to restrict CHIKV and DENV infection in vivo.I do not have any problem with the data, but I do have two major questions that center on methods and the Discussion.1) In SiglecH IRF7/IRF7 mice, expression of IRF7 is not controlled by the IRF7 promoter. SiglecH promoter-controlled expression of IRF7 in pDCs may have very distinct effects from regulated expression of IRF7 by the IRF7 promoter. The manuscript would be strengthened if the authors could discuss limitations associated with this particular issue.

We thank the reviewer for the suggestion. This aspect is now discussed see text:

“In this mouse model, the restoration of Irf7 expression in pDCs does not reflect control under the endogenous promoter, as the Irf7 gene is under the control of the pDC-specific Siglech promoter. […] Furthermore, Siglech expression by pDCs is inversely correlated to ISG upregulation in vivo (Figure 4—figure supplement 1E), in agreement with prior study (Puttur et al., 2013).”

2) If I understood the methods correctly, mice of different genotypes appear to have been infected at an age range of up to 3 weeks. This difference in ages can make a substantial impact on vulnerability to viral infections (for example, during West Nile virus infection, infection at age 5 weeks results in ~90% lethality, whereas infection at 8 weeks results in ~75% lethality.) Instead of providing a range of ages, could the authors be more precise about the age of infected mice of each genotype?

We now provide a table with the list of the age and sex of mice used for all the experiments, as well as the number of individual cohorts used for each genotype/time point in these studies (new Supplementary file 1). Mice of different genotypes within cohorts (or across cohorts when those comparisons were performed) were within 3 weeks of age of each other. We did not observe any apparent differences according to the age of the mice.

Reviewer #2:The manuscript by Webster et al. describes the assessment of pDCs in two different arboviruses: dengue (DENV) and chikungunya (CHIKV) through the role of IRF7-driven IFN production. In general, the study is technically well-conducted and to decipher the immunomodulatory functions of pDCs in arbovirus sensing, authors generated the pDC:Irf7^+^ mouse model in C57BL/6 mice in order to restrict the role of IRF7 to pDCs. A wealth of data was produced that showed an indirect role for pDC IRF7-signalling in both viruses. However, the authors should consider removing one virus to streamline the datasets to improve clarity. As it stand, the datasets are scattered and all over the place due to the different viral characteristics of the 2 viruses. Specific comments below:1) Introduction – although well-written, the Introduction is too long and clumsy. This is expected as due to the description of 2 viruses. Both are different with specific characteristics. It could be significantly shortened and more focused.

As above-mentioned in the response to the Editor, the text and figures were considerably reorganized, focusing first on dengue virus and with the key findings on CHIKV infection separated into a new Figure 7 and related new Figure 7—figure supplement 1. The Introduction has been focused accordingly.

2) Figures 1 and 2 on cell-cell sensing of DENV and CHIKV – the infection conditions and viral kinetics between these 2 viruses are very different. Where infections conducted using the same batch of cells? Nonetheless, have the authors done a direct comparison between DENV and CHIKV in terms of fold change in IFNa levels? At 24h, 1 out of 3 replicates had TNFa levels of closed to 100 pg/ml. Is this sufficient to conclude that there was minimal or no TNFa production? In general, there are insufficient replicates with no statistics done.

The reviewer is correct that some infection conditions and viral kinetics are different between DENV/CHIKV. For example, BHK21 cells were used as DENV-infected cells (Figure 1A-B) and Huh7.5.1 cells were used as CHIKV-infected cells (Figure 7E) in cocultures with human pDCs. However, the xenogeneic nature of the DENV cocultures was purposeful, to allow qPCR analysis specifically of the human pDCs rather than DENVinfected cells by using species-specific primer sets (Figure 1—figure supplement 1D). Importantly, this manuscript and prior work (Decembre et al., 2014) have demonstrated that pDC responses are robust and not dependent on specific coculture conditions (e.g., tissue origin and/or species of the virally-infected cells).

Additionally, as suggested by this reviewer, we performed a direct comparison of pDC response to cells infected by either DENV or CHIKV. As shown in Author response image 1, the IFNα levels are within the same range.

**Author response image 1. respfig1:** Quantification of IFNα by ELISA in supernatants of human pDCs cocultured with Huh7.5.1 cells infected by either CHIKV or DENV, or uninfected. Cells were infected 48 hours prior to cocultures.

Further, we performed new additional experiments and statistical analyses for ex vivo studies of TNFα versus IFNα production as well as other ex vivo studies, now included in:

- new Figure 1 panels A and B

- new Figure 3 panels A and B

These panels include additional new results.

For the comparison of pDC cytokine production across mouse genotypes, rank-order statistical analysis (e.g., Kruskal-Wallis) was necessary due to many non-parametric data points at the limit of detection of the ELISA assay. Strong statistically significant differences were observed when comparing IFNα secretion between only pDC:IRF7^+^ and Irf3^-/-^/7^-/-^ pDCs in response to various TLR7/8 stimuli. However, the inclusion of WT pDCs, with greater IFNα secretion in these analyses, distorted the rank orders in a fashion that very large sample sizes would be necessary to ascertain statistical differences between pDC:IRF7^+^ and Irf3^-/-^/7^-/-^ pDCs (n>8 independent pDC isolations). We believe it is more informative to include data from WT pDCs in these analyses, even at the expense of losing statistical power for the pDC:IRF7^+^ versus Irf3^-/-^/7^-/-^ comparison, especially as the pattern of IFNα secretion was broadly similar across a range of TLR7/8 stimuli (DENV, Flu, IMQ, R848).

We have now included new Figure 7 panels A-to-E.

3) Figures 3 to 7 on in vivo responses of DENV and CHIKV in pDC:Irf7^+^ mice – a mix of DENV and CHIKV datasets are distributed between the main and supplementary figures. Insufficient data points for proper statistics to be performed that weakened the study. Have the authors considered putting the levels of ISGs from 2 different types of infections together as a comparative? This will be more insightful to understand the similar and different mechanistic effects of virus sensing on IRF signalling on pDCs. Heat-maps provided in Figure 4 and Figure 3—figure supplement 1 suggest different signatures in terms of fold change. Otherwise, the rationale of combining 2 viruses into 1 story is not compelling.

As above-mentioned, the text and figures are now deeply reorganized to separate DENV and CHIKV. In vivo results now include additional data and statistical analysis:

- in new Figure 2 panels B and C, with inclusion of new data;

- in new Figure 6 panels B-to-E and corresponding Figure 6—figure supplement 2, with inclusion of new mouse cohort.

The comparison for ISGs would be interesting but the study for CHIKV infection was not designed for this type of analysis, so none of these samples or data are available. We agree that, moving forward, extending these types of analyses of IFN signaling to viruses other than DENV will be quite interesting in the future.

The heat-maps from new Figure 3E-G (previous Figure 4) and Figure 3—figure supplement 1 are indeed presented differently: results in new Figure 3E-G are expressed as Z-score and are magnitude-independent while the fold change in Figure 3—figure supplement 1 provides information on the magnitude of the change. The reasoning for these distinct displays was to show the general pattern of ISG/NF-κB gene change in Figure 3E-G, while providing the data on the magnitude of the response for those interested in Figure 3—figure supplement 1. This has been clarified in the text, as follows: “(Figure 3E-G, expressed as magnitude-independent Z score, and Figure 3—figure supplement 1, expressed as magnitude-dependent fold-change)”.